# PreBötzinger complex neurons drive respiratory modulation of blood pressure and heart rate

Clément Menuet[1,2]*, Angela A Connelly[1], Jaspreet K Bassi[1], Mariana R Melo[1], Sheng Le[3], Jessica Kamar[1], Natasha N Kumar[4], Stuart J McDougall[5], Simon McMullan[3], Andrew M Allen[1,5]*

[1]Department of Physiology, University of Melbourne, Victoria, Australia; [2]Institut de Neurobiologie de la Méditerranée, INMED UMR1249, INSERM, Aix-Marseille Université, Marseille, France; [3]Faculty of Medicine & Health Sciences, Macquarie University, NSW, Australia; [4]Department of Pharmacology, School of Medical Sciences, University of New South Wales, NSW, Australia; [5]Florey Institute of Neuroscience and Mental Health, University of Melbourne, Victoria, Australia

**Abstract** Heart rate and blood pressure oscillate in phase with respiratory activity. A component of these oscillations is generated centrally, with respiratory neurons entraining the activity of presympathetic and parasympathetic cardiovascular neurons. Using a combination of optogenetic inhibition and excitation in vivo and in situ in rats, as well as neuronal tracing, we demonstrate that preBötzinger Complex (preBötC) neurons, which form the kernel for inspiratory rhythm generation, directly modulate cardiovascular activity. Specifically, inhibitory preBötC neurons modulate cardiac parasympathetic neuron activity whilst excitatory preBötC neurons modulate sympathetic vasomotor neuron activity, generating heart rate and blood pressure oscillations in phase with respiration. Our data reveal yet more functions entrained to the activity of the preBötC, with a role in generating cardiorespiratory oscillations. The findings have implications for cardiovascular pathologies, such as hypertension and heart failure, where respiratory entrainment of heart rate is diminished and respiratory entrainment of blood pressure exaggerated.

*For correspondence:
clement.menuet@inserm.fr (CéM);
a.allen@unimelb.edu.au (AMA)

**Competing interests:** The authors declare that no competing interests exist.

## Introduction

The respiratory and cardiovascular systems act in synergy to maintain blood gas homeostasis in animals. Beyond their respective separate roles, these systems are also functionally coupled, with oscillations in blood pressure (BP), called Traube-Hering waves, and in heart rate (HR), called respiratory sinus arrhythmia (RSA), that are in phase with the respiratory cycle. This respiratory entrainment of cardiovascular activity is a highly conserved physiological property present in vertebrates, including mammals, fish, amphibians and reptiles (*Taylor et al., 2010*; *Taylor et al., 1999*), and was recently found in the early air-breathing primitive lungfish (*Monteiro et al., 2018*). When within physiological range, Traube-Hering waves are important contributors to vascular tone changes (*Briant et al., 2015*), RSA reduces cardiac energetic cost (*Ben-Tal et al., 2012*), and both are believed to optimize blood tissue perfusion and gas exchange. However, an alteration in the amplitude of these oscillations, typically with enlargement of Traube-Hering waves and concurrent abolishment of RSA, occurs in common cardiovascular diseases (e.g. hypertension and heart failure) and contributes to their pathogeneses (*Machado et al., 2017*; *Menuet et al., 2017*; *Palatini and Julius, 2009*; *Task Force of the European Society of Cardiology and the North American Society of Pacing and Electrophysiology, 1996*). The pathological relevance of these oscillations extends to numerous

neurological diseases, not superficially linked to cardiovascular function, including depression (*Brush et al., 2019*) and amyotrophic lateral sclerosis (*Merico and Cavinato, 2011*).

Traube-Hering waves and RSA are generated by multiple mechanisms, including changes in thoracic pressure and viscerosensory afferent feedback, but their origin has a central component, as they remain in heart- and lung-denervated humans, and in paralyzed and deafferented animal preparations (*Dick et al., 2014*; *Farmer et al., 2016*; *Menuet et al., 2017*; *Taha et al., 1995*). Traube-Hering waves are generated by bursts of vasomotor sympathetic nerve activity that occur in phase with the respiratory cycle (RespSNA), spanning most of inspiration and the very beginning of post-inspiration (the first phase of expiration). As a result of the slow vascular contraction dynamics, Traube-Hering waves peak during post-inspiration. RespSNA is the result of respiratory modulation of the activity of pre-sympathetic neurons in the rostral ventrolateral medulla oblongata, and in particular of the excitatory inspiratory modulation of adrenergic C1 neurons (*Menuet et al., 2017*; *Moraes et al., 2013*). RSA is generated by a cardio-inhibitory vagal parasympathetic tone that decreases during the second part of expiration and into inspiration, resulting in HR oscillations that peak at the end of inspiration. Cardiac vagal preganglionic neurons of the nucleus ambiguus are weakly inhibited during the second part of expiration, and strongly inhibited during inspiration, by both GABAergic and glycinergic inputs (*Gilbey et al., 1984*; *Neff et al., 2003*). However, the specific cellular components of the central respiratory pattern generating circuit that are responsible for modulation of both vasomotor pre-sympathetic and cardiac vagal preganglionic neurons are not fully elucidated.

Respiratory and cardiovascular neurons are present in adjacent and slightly intermingled cell columns in the ventral medulla oblongata. The respiratory neurons are classified by the relationship between their activity pattern and phase of the respiratory cycle – for example inspiratory neurons fire action potentials during inspiration. Core to the respiratory column is a small, yet heterogeneous, group of neurons in the preBötzinger Complex (preBötC), which consists of excitatory (glutamatergic) inspiratory neurons, and inhibitory (GABAergic and glycinergic) neurons that can be inspiratory or expiratory (*Baertsch et al., 2018*; *Koizumi et al., 2013*; *Morgado-Valle et al., 2010*; *Sherman et al., 2015*). The preBötC is critical for respiratory rhythm generation, and also acts as a master oscillator entraining the activity of other oscillators, like the sniffing and whisking oscillators, in a unidirectional fashion (*Deschênes et al., 2016*; *Moore et al., 2013*). Our recent work on the pre-synaptic connectome of C1 neurons shows that preBötC neurons have the anatomical connections to provide the inspiratory input to these pre-sympathetic neurons (*Dempsey et al., 2017*; *Menuet et al., 2017*). It is also proposed that the strong inspiratory inhibition of cardiac vagal preganglionic neurons comes from the preBötC (*Frank et al., 2009*).

Using optogenetics, which provides the spatial resolution to restrict opsin expression to a small group of neurons such as those in the preBötC, and temporal control to excite or inhibit neurons during specific phases of the respiratory cycle, we show that the preBötC is a cardiorespiratory hub that drives RespSNA, Traube-Hering waves and RSA.

## Results

### Injections of lentivirus expressing the light-activated anion channel *Gt*ACR2 are targeted to the preBötC

To induce rapid, reversible and powerful inhibition of neural activity in the preBötC, we injected a lentivirus inducing expression of *Gt*ACR2 (*Govorunova et al., 2015*) bilaterally. Injections were directed by electrophysiological recordings to identify inspiratory neurons in the rostral-most part of the ventral inspiratory cell column in the medulla oblongata. *Post hoc* histological analysis revealed the *Gt*ACR2 expression was targeted to the gap in the rostrocaudal column of parvalbumin-expressing neurons (*Figure 1A*; *Alheid et al., 2002*), dorsal to tyrosine hydroxylase (TH)-expressing A1-C1 neurons, ventral to the caudal tip of the compact part of the nucleus ambiguus, and at the level containing the highest density of neurokinin type 1 receptor (NK$_1$R)-positive neurons (*Figure 1B*). Together these markers characterize the preBötC area. *Gt*ACR2 was expressed in both excitatory (glutamatergic) and inhibitory (GABAergic and glycinergic) neurons in the preBötC (*Figure 1C–E*). A heatmap representation of *Gt*ACR2 expression in all the 12 animals used for the in situ Working

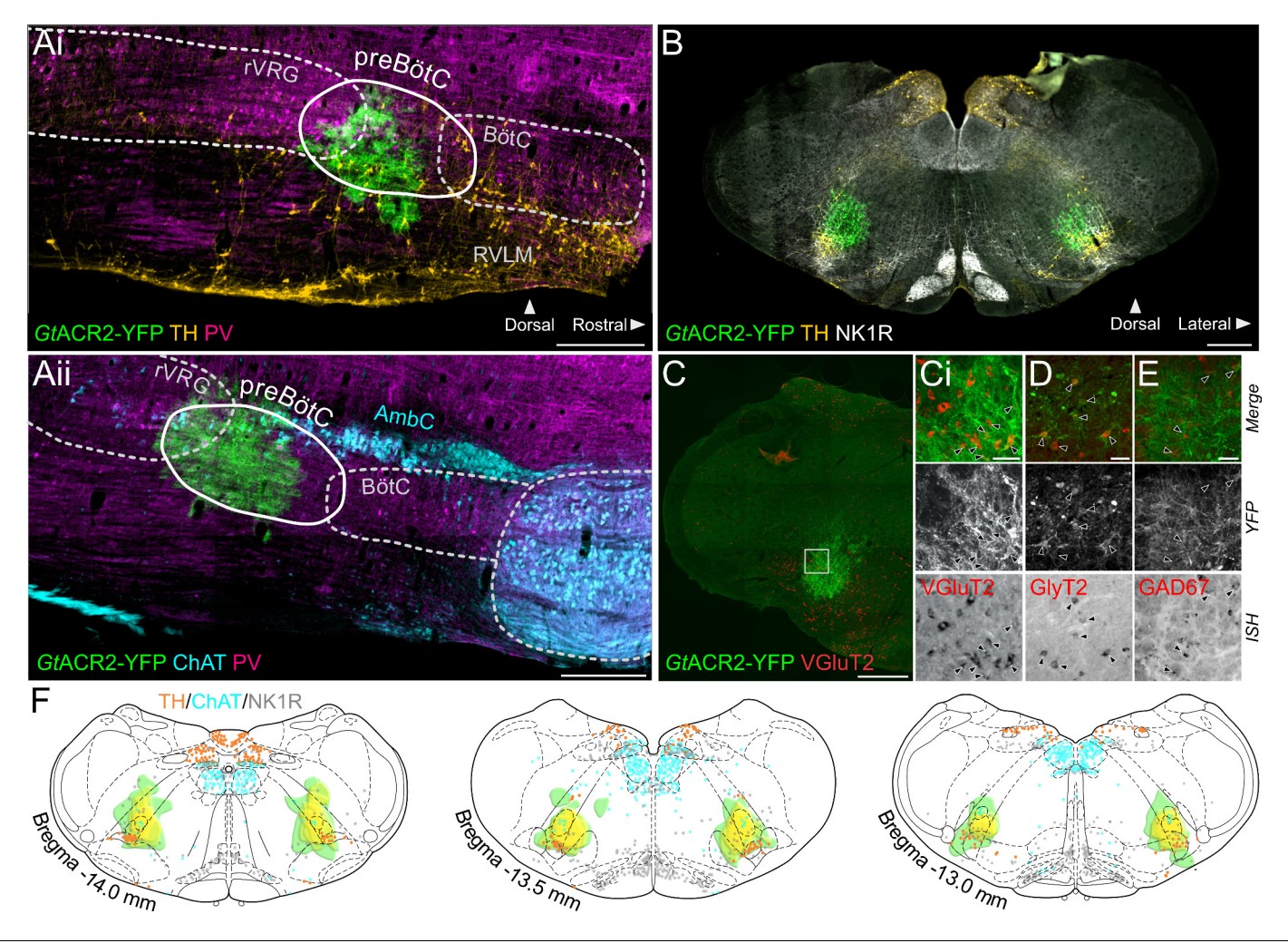

**Figure 1.** Expression of *Gt*ACR2, in the preBötC. (A) Sagittal and (B) coronal sections showing expression of *Gt*ACR2-yellow fluorescent protein (YFP) and parvalbumin (PV), tyrosine hydroxylase (TH), choline acetyltransferase (ChAT) and neurokinin type 1 receptor (NK1R) immunoreactivity. (C) In situ hybridization showing *Gt*ACR2-YFP expression in glutamatergic (Ci), glycinergic (D) and GABAergic (E) neurons in the preBötC, as indicated by black arrows. (F) Schematic coronal maps, based on atlas of *Paxinos and Watson, 2004*, showing the distribution of *Gt*ACR2-YFP from all rats (n = 12) used in Working Heart-Brainstem Preparation experiments relative to TH, ChAT and NK1R expression. Each rat is depicted in green with the gradation to yellow indicating increasing overlap in expression between animals, and indicating that the common expression area overlies the preBötC. Scale bars are 500 μm (Ai, Aii, B, C) or 50 μm (Ci, D, E).

Heart-Brainstem Preparation (WHBP) experiments shows that intersectional *Gt*ACR2 expression across all animals is restricted to the preBötC area (*Figure 1F*).

## The preBötC is necessary for inspiratory rhythm generation

The preBötC is mostly studied for its critical role in inspiratory rhythm generation. Prolonged *Gt*ACR2-mediated preBötC photoinhibition, either in vivo in anesthetized rats (n = 5) or in the in situ WHBP (n = 12), produced immediate cessation of inspiratory activity and a long-lasting apnea (*Figure 2A–C*; *Figure 3A–C*). Following the apnea, whilst preBötC photoinhibition remained activated, inspiratory activity resumed, but with an altered pattern of intercostal muscle (intC) electromyograph (EMG) activity in vivo (*Figure 2A–B*) or phrenic nerve activity in situ (PNA; *Figure 3A–C and H*; *Figure 3—figure supplement 1A–B*). In the WHBP, the longest apnea was produced by 10 and 20 Hz photoinhibition (*Figure 3A–C*). When PNA resumed, 5–10 Hz photoinhibition induced shorter and smaller PNA bursts, whilst PNA burst characteristics were not altered by 20–50 Hz photoinhibition.

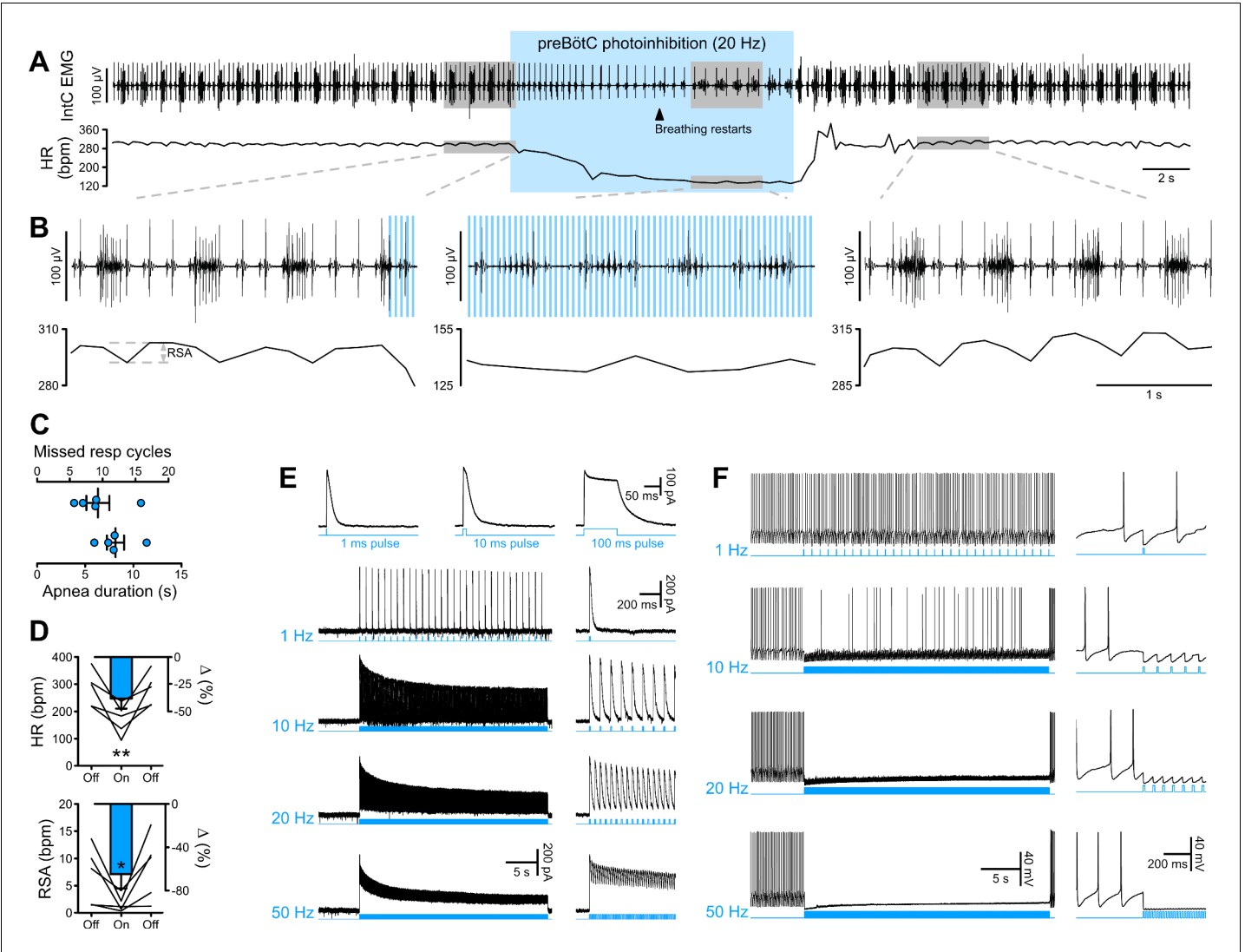

**Figure 2.** PreBötC photoinhibition in vivo induces respiratory cessation, bradycardia, and suppresses RSA, and *Gt*ACR2 photoactivation in vitro causes strong neuronal inhibition. (**A**) Intercostal (intC) electromyograph (EMG) recording in an anesthetized P31 rat with bilateral *Gt*ACR2 expression in the preBötC, showing bursts of inspiratory activity with the electrocardiogram superimposed, from which heart rate (HR) is derived. Prolonged (20 Hz, 10 ms pulses, 300 pulses) preBötC photoinhibition, starting during inspiration, immediately stopped the inspiratory burst and induced a long apnea, before respiratory activity resumed even though preBötC photoinhibition continued. PreBötC photoinhibition also induced a profound bradycardia, and a decrease in arterial pressure as shown on *Figure 2—figure supplement 1* on a separate animal cohort (n = 3). (**B**) Enlargements of the shaded areas in (**A**), showing that the first pulse of light immediately stops the intC EMG inspiratory burst, and that when breathing activity resumes during preBötC photoinhibition the intC EMG inspiratory bursts are crenelated with periodic cessations of intC EMG activity after each light pulse. Note that the overall intC EMG ramping discharge pattern is maintained. Respiratory sinus arrhythmia (RSA), the oscillations in HR in phase with respiratory activity, is abolished during preBötC photoinhibition, even when breathing activity resumes. (**C–D**) PreBötC photoinhibition induced a prolonged cessation of breathing activity, bradycardia and suppression of RSA in all animals tested (n = 5, mean ± SEM for group values; one-way repeated measures ANOVA followed with post hoc Holm-Sidak multiple-comparison test performed on raw data, as shown on the associated source data and detailed statistics; *p<0.05, **p<0.01, photoinhibition vs. control and recovery conditions). (**E–F**) Whole-cell recordings (low Cl⁻ internal solution, ECl⁻=−68.8 mV) of *Gt*ACR2-positive solitary tract nucleus neurons in brainstem slices in voltage clamp ((**E**), holding voltage −40 mV) and current clamp (**F**) modes. (**E**) *Gt*ACR2 photostimulation induced large inhibitory conductances that showed fast activation and deactivation kinetics, and little inactivation, as previously shown (*Govorunova et al., 2015*). The photocurrents persisted throughout the prolonged photostimulation protocols. (**F**) In a spontaneously active cell, *Gt*ACR2 photostimulation prevented action potential firing when the light pulse was delivered at the appropriate timing, resulting in complete firing cessation during prolonged high frequency photostimulations including and above 20 Hz.

The online version of this article includes the following source data and figure supplement(s) for figure 2:

**Source data 1.** Source data and statistics for *Figure 2*.

**Figure supplement 1.** PreBötC photoinhibition in vivo induces respiratory cessation, bradycardia, and decreases arterial pressure.

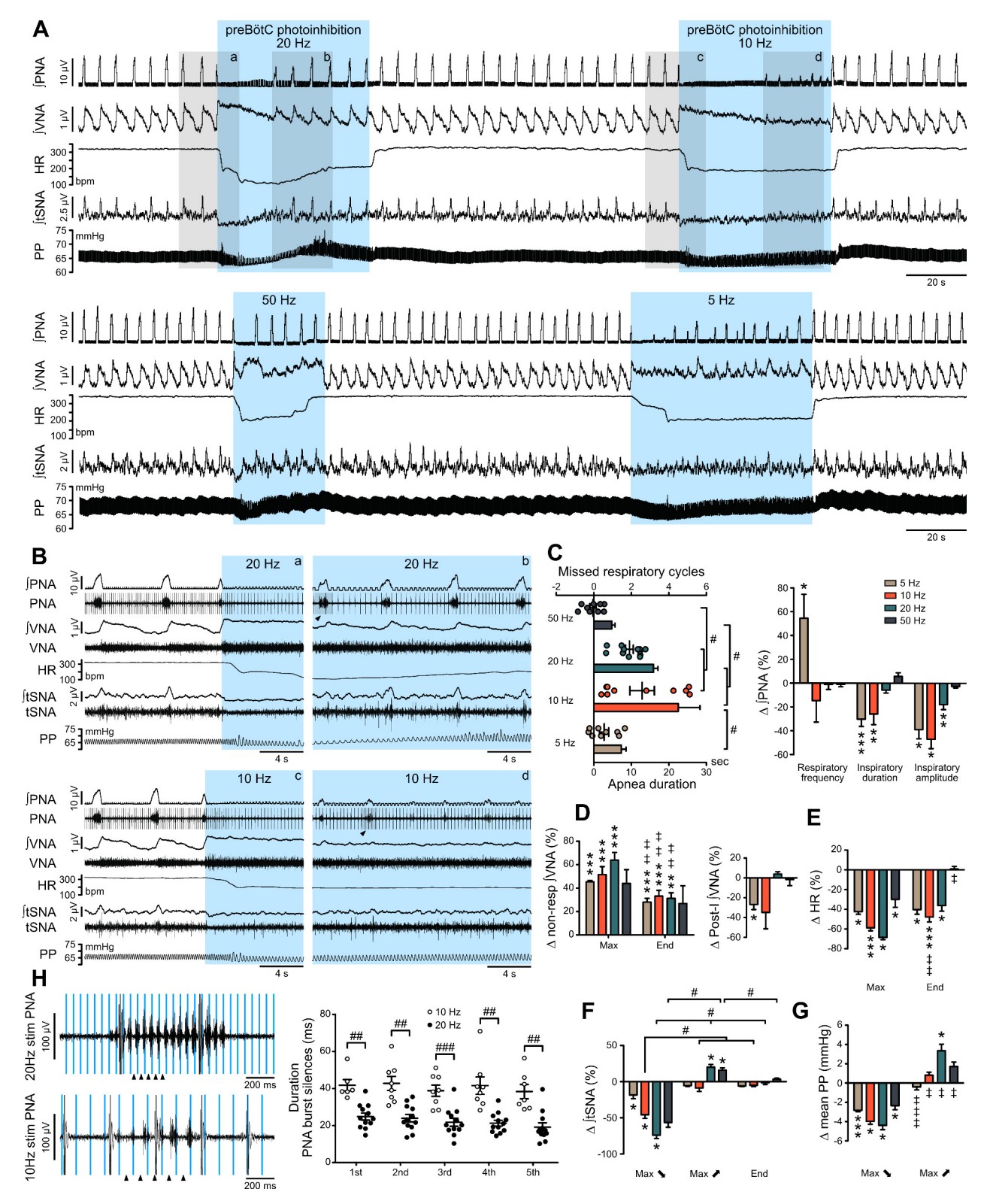

**Figure 3.** Effect of prolonged preBötC photoinhibition in situ on cardio-respiratory function. (**A**) Recording of phrenic nerve activity (PNA), vagus nerve activity (VNA) and thoracic sympathetic nerve activity (tSNA), heart rate (HR) and perfusion pressure (PP), in an in situ Working Heart-Brainstem Preparation (WHBP) with bilateral photoinhibition of the preBötC at 5, 10, 20 and 50 Hz. Following the initial apnea, PNA resumed despite continued preBötC photoinhibition. VNA increased and HR decreased for the entire photoinhibition period. A transient decrease in tSNA and PP, concomittent

*Figure 3 continued on next page*

*Figure 3 continued*

with the apnea, was followed by increases in tSNA and PP with the return of PNA. (B) Higher resolution recordings of the shaded areas in (A) showing the strongest physiological effects occurred with 20 Hz and 10 Hz photoinhibition. (C–G) Group data (n = 12), expressed as change relative to the pre-photoinhibition control period, examining changes in respiratory parameters, including PNA characteristics following resumption of activity during continued photoinhibition (C); VNA measured outside of inspiratory/post-inspiratory activities (non-resp) and in the post-inspiratory period (D); HR (E) tSNA (F) and PP (G). Details regarding VNA and tSNA analysis are shown on *Figure 3—figure supplement 2*. (H) Higher temporal resolution recordings of the periods denoted by arrowheads in (B), showing the crenelated PNA burst shape, yet with maintained ramping discharge pattern. Analysis of the first five PNA burst silence periods, denoted by arrowheads in (H), showed the 20 Hz photoinhibition induced shorter silence durations. Group data are presented as mean ± SEM; Friedman repeated measures analysis of variance on ranks followed with pairwise multiple comparison Tukey test, one-way repeated measures ANOVA followed with *post hoc* Holm-Sidak multiple-comparison test, paired student's t-test or Wilcoxon signed rank test, performed on raw data, as shown on *Figure 3—figure supplement 1* and associated source data and detailed statistics; *p<0.05, **p<0.01, ***p<0.001, photoinhibition vs. control and recovery conditions; # p<0.05, ## p<0.01, ### p<0.001, between compared data; ‡ p<0.05, ‡‡ p<0.01, ‡‡‡ p<0.001, intra-photostimulation frequency vs. max change.

The online version of this article includes the following source data and figure supplement(s) for figure 3:

**Source data 1.** Source data and statistics for *Figure 3*.

**Figure supplement 1.** Individual data showing alterations of respiratory, sympathetic vasomotor and cardiac parasympathetic activities during prolonged *Gt*ACR2-mediated preBötC photoinhibition in situ.

**Figure supplement 2.** Method for analysis of the different components of VNA and tSNA.

A single 10 ms light pulse at any time during inspiration was sufficient to immediately stop the PNA burst and trigger post-inspiration (Figure 7A–C). The duration of the following expiratory period was correlated to the duration and amplitude of the stopped inspiration, mostly by adjusting the post-inspiratory phase duration. Photoinhibition of the PreBötC only during the expiratory period prolonged the expiratory time for the duration of the photoinhibition, and PNA activity resumed following a short (~1 s) post-photoinhibition delay (Figure 7E).

In whole-cell recordings of *Gt*ACR2-expressing neurons (n = 7) in brainstem slices of the dorsal medulla, we observed strong, and sustained, hyperpolarisation with *Gt*ACR2 photoactivation (*Figure 2F*), induced by large inhibitory conductances (*Figure 2E*). This hyperpolarisation continued for the entire period of light delivery, and in spontaneously active neurons, complete blockade of action potential firing was obtained with 10 ms duration light pulses at 20 Hz. On the basis of these recordings, it is not clear why inspiratory activity resumed during prolonged photoinhibition.

## Prolonged preBötC photoinhibition induces decreased vasomotor sympathetic activity and increased cardiac parasympathetic activity

Due to the profound apnea, which would alter blood gas composition and sensory afferent input, detailed analysis of the cardiorespiratory response to preBötC photoinhibition was not feasible in vivo and hence detailed analyses were performed in the WHBP, an oxygenated, perfused preparation devoid of the depressive effects of anaesthesia and where central respiratory perturbations do not influence peripheral blood gases or pH (*Menuet et al., 2017*; *Paton, 1996*). During prolonged preBötC photoinhibition, thoracic sympathetic nerve activity (tSNA) showed initial inhibition during apnea, and then a return toward control levels as central respiratory activity resumed (*Figure 3A and F*; *Figure 3—figure supplement 1E and G*; *Figure 3—figure supplement 2A*). This was reflected in an initial decrease in perfusion pressure (PP), followed by an acute increase (*Figure 3A and G*; *Figure 3—figure supplement 1F–G*) that were maximal with 20 Hz photoinhibition. We also observed a strong increase in vagal nerve activity (VNA) and decrease in HR (bradycardia) (*Figure 2A–B and D*; *Figure 3A–B and D–E*; *Figure 3—figure supplement 1C–D*). Unlike the sympathetic vasomotor effect, the bradycardia lasted the entire photoinhibition period and was not correlated to the apnea duration (*Figure 3—figure supplement 1G*). The bradycardia was due to increased cardiac parasympathetic drive (measured as non-resp VNA, *Figure 3A and D*, *Figure 3—figure supplement 2B*) as it was abolished by bath application of the muscarinic antagonist, atropine (*Figure 4*). The biphasic tSNA and PP responses to preBötC photoinhibition were maintained during atropine (*Figure 4C*).

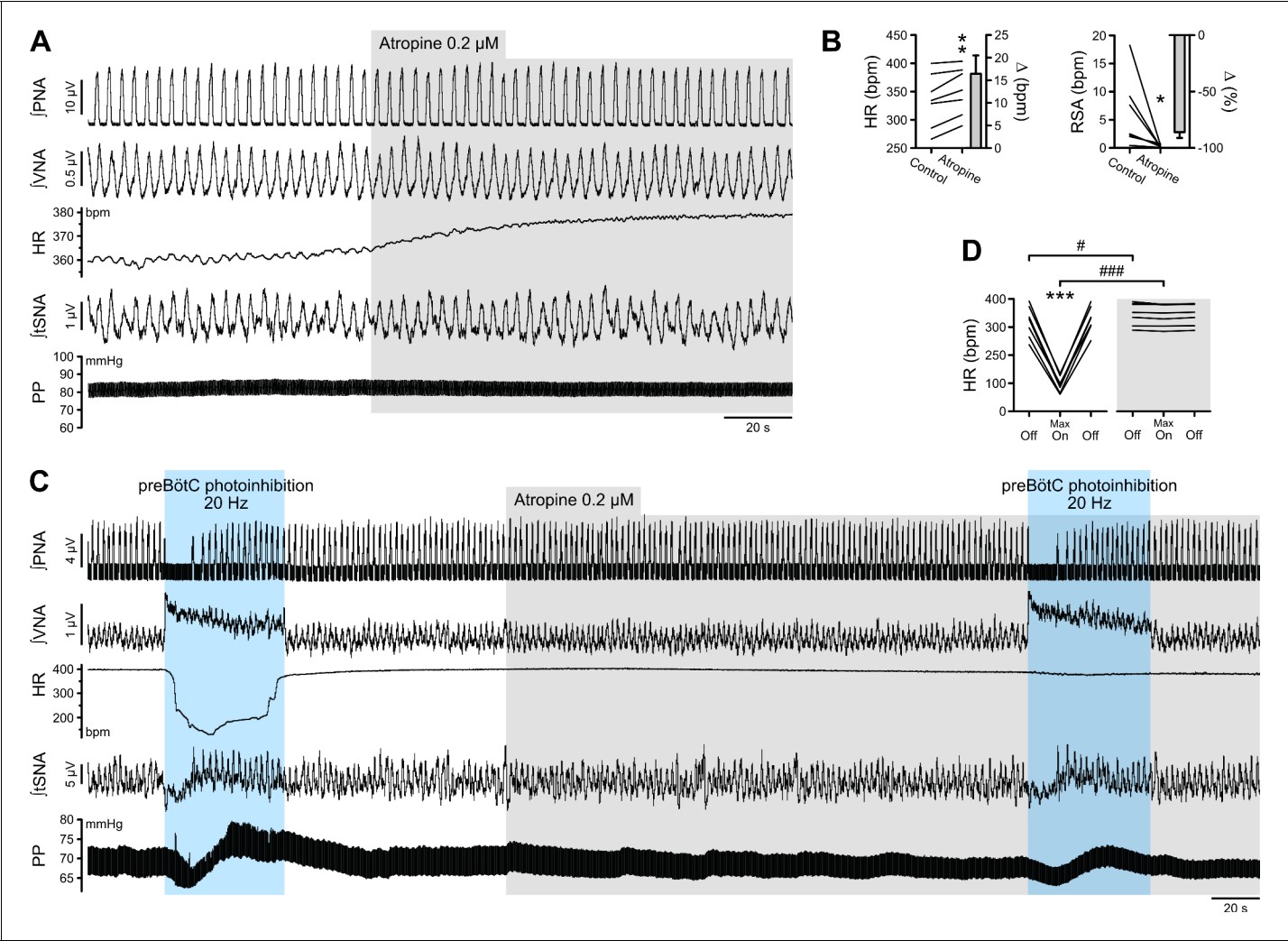

**Figure 4.** The bradycardia induced by preBötC photoinhibition is mediated by increased cardiac parasympathetic drive. (**A**) The effect of bath application of the muscarinic antagonist, atropine, in the in situ WHBP of a P31 rat. (**B**) Group data (n = 7) showing that atropine induced an increase in HR and abolished RSA, confirming its effective blockade of cardiac parasympathetic modulation. (**C**) Prolonged 20 Hz preBötC photoinhibition before, and after, bath application of atropine. (**D**) Atropine did not affect the increase in VNA induced by preBötC photoinhibition, but totally blocked the decrease in HR, showing that preBötC photoinhibition induces increased cardiac parasympathetic drive. The sympathetic vasomotor effects induced by preBötC photoinhibition were not altered by atropine, showing that they are not mediated by baroreflex afferents. Group data are presented as mean ± SEM; paired student's *t* test, Wilcoxon signed rank test or two-way repeated measures ANOVA followed with post hoc Holm-Sidak multiple-comparison test performed on raw data, as shown on the associated source data and detailed statistics; *p<0.05, **p<0.01, ***p<0.001, photoinhibition vs. control condition (**B**), or vs. control and recovery conditions (**D**); # p<0.05, ### p<0.001, between photostimulation conditions.

The online version of this article includes the following source data for figure 4:

**Source data 1.** Source data and statistics for *Figure 4*.

## The preBötC is sympatho-excitatory and parasympatho-inhibitory

To avoid secondary circuit effects arising from sustained photoinhibition, we used single light-pulse preBötC photoinhibition to test direct functional connectivity with sympathetic and parasympathetic activities. Laser-triggered averaging of tSNA and VNA following low-frequency (1 Hz) preBötC photoinhibition confirmed opposite sympathetic and parasympathetic effects (*Figure 5*; *Figure 5—figure supplement 1*). PreBötC photoinhibition induced a small sympatho-excitation that peaked ~65 ms following the light pulse, and mainly a strong sympatho-inhibition that was maximal at ~135 ms. PreBötC photoinhibition induced dramatic parasympathetic excitation, which could be seen on VNA recordings in response to individual light pulses (*Figure 5A–B*). This parasympatho-excitation was very fast, peaking at ~7 ms (*Figure 5C*).

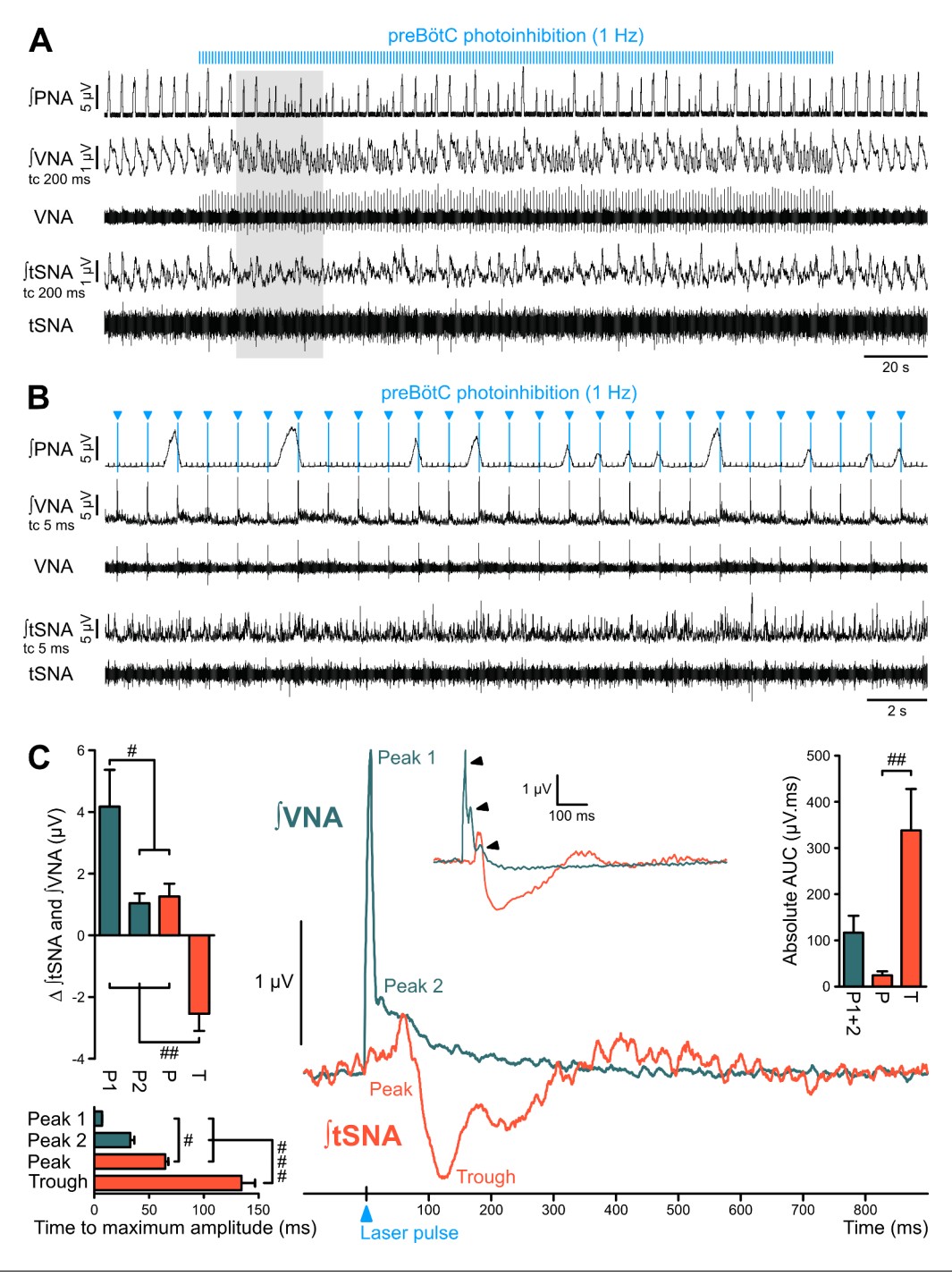

**Figure 5.** The preBötC is sympatho-excitatory and parasympatho-inhibitory. Traces of activity in the WHBP at low (**A**) and higher (**B**) time resolution, showing that preBötC photoinhibition at 1 Hz disrupts PNA, VNA and tSNA. (**C**) Laser triggered averages of integrated VNA and tSNA (tc 5 ms, 300 pulses), show that preBötC photoinhibition induced: a rapid excitation of VNA at short latency, sometimes accompanied by a second or third peak (arrowheads in inset); and an initial small tSNA peak and a subsequent larger tSNA trough of longer latency. These effects were found irrespective of the respiratory phase at which the light pulse was delivered. Group data are presented as mean ± SEM; one-way repeated measures ANOVA followed with *post hoc* Holm-Sidak multiple-comparison test performed on raw data, as shown on *Figure 5—figure supplement 1* and associated source data and detailed statistics; # $p<0.05$, ## $p<0.01$, ### $p<0.001$, between respective effects.

The online version of this article includes the following source data and figure supplement(s) for figure 5:

*Figure 5 continued on next page*

*Figure 5 continued*

**Source data 1.** Source data and statistics for *Figure 5*.
**Figure supplement 1.** Individual data showing sympathetic and parasympathetic effects of 1 Hz preBötC photoinhibition.

We conclude that neurons within the preBötC exert direct excitatory effects on vasomotor sympathetic pathways and inhibitory effects on cardiac parasympathetic pathways. We next investigated the role played by PreBötC neurons in the respiratory modulation of tSNA (RespSNA), PP (Traube-Hering waves) and HR (RSA).

## PreBötC activity contributes to the generation of RespSNA and Traube-Hering waves

When respiration resumed during prolonged preBötC photoinhibition, the return of RespSNA and Traube-Hering waves mirrored the return of PNA inspiratory activity (*Figure 3A–C* and *Figure 6A*). Like PNA, both RespSNA and Traube-Hering wave amplitudes were strongly decreased during 10 Hz preBötC photoinhibition, but unaffected during 20 Hz photoinhibition (*Figure 6A–B*; *Figure 6—figure supplement 1A* and 6S1D), and overall individual changes in RespSNA area under the curve (AUC) and Traube-Hering wave amplitude correlated closely with changes in PNA amplitude (*Figure 6C*). Decreased RespSNA duration during 10 Hz photoinhibition was specifically due to a shorter inspiratory component (*Figure 6D*; *Figure 6—figure supplement 1A and C*). In contrast, mean tonic tSNA was not altered during preBötC photoinhibition (*Figure 6B*; *Figure 6—figure supplement 1B*).

Single light pulses delivered during early inspiration arrested PNA bursts (AUC reduced by ~75%) and almost abolished the associated RespSNA and Traube-Hering wave (*Figure 7A–D*; *Figure 7—figure supplement 1A*). PreBötC photoinhibition during early post-inspiration had no impact on PNA or RespSNA and induced a small increase in Traube-Hering wave amplitude. Tetanic preBötC photoinhibition during expiration induced exaggerated RespSNA and Traube-Hering wave amplitude, which is likely due to the increased pulse pressure associated with the bradycardia (*Figure 7E*).

Together, these results show that the preBötC exerts a direct phasic inspiratory drive to vasomotor sympathetic activity, with a major involvement in the generation of RespSNA and Traube-Hering waves.

## PreBötC activity is strongly involved in the generation of RSA

RSA was almost abolished during both 10 Hz and 20 Hz preBötC photoinhibition (*Figure 6A–B*; *Figure 6—figure supplement 1D*), and changes in RSA were independent from changes in PNA amplitude (*Figure 6C*). The loss of baseline RSA evoked by systemic atropine demonstrates that RSA is generated *via* parasympathetic activity (*Figure 4A–B*). These results indicate that the preBötC is a source of ongoing respiratory phase-locked inhibitory input to cardiac parasympathetic preganglionic neurons.

Baseline RSA is characterized by peaks in HR at the end of inspiration, decreases during the first part of post-inspiration, and increases continuously during the second part of post-inspiration, late-expiration (E2) and inspiration (*Figure 7—figure supplement 1B*). Single light pulse preBötC photoinhibition during inspiration did not affect the amplitude of the RSA during the respiratory cycle in which stimulation occurred (*Figure 7A–D*; *Figure 7—figure supplement 1*), but rather decreased RSA amplitude in the following respiratory cycle due to a smaller post-photoinhibition HR decrease that didn't reach baseline level before HR increased again. Photoinhibition of preBötC during early post-inspiration induced a larger decrease in HR which increased the amplitude of the following cycle of RSA. Prolonged photoinhibition during expiration strongly increased VNA, decreased HR and prevented the tachycardic component of RSA until photoinhibition was stopped, whereupon HR increased abruptly prior to the next inspiration (*Figure 7E*).

Together, these results show that the preBötC exerts a complex cardiac parasympathetic influence that spans both inspiratory and expiratory periods, with a major involvement in the generation of RSA.

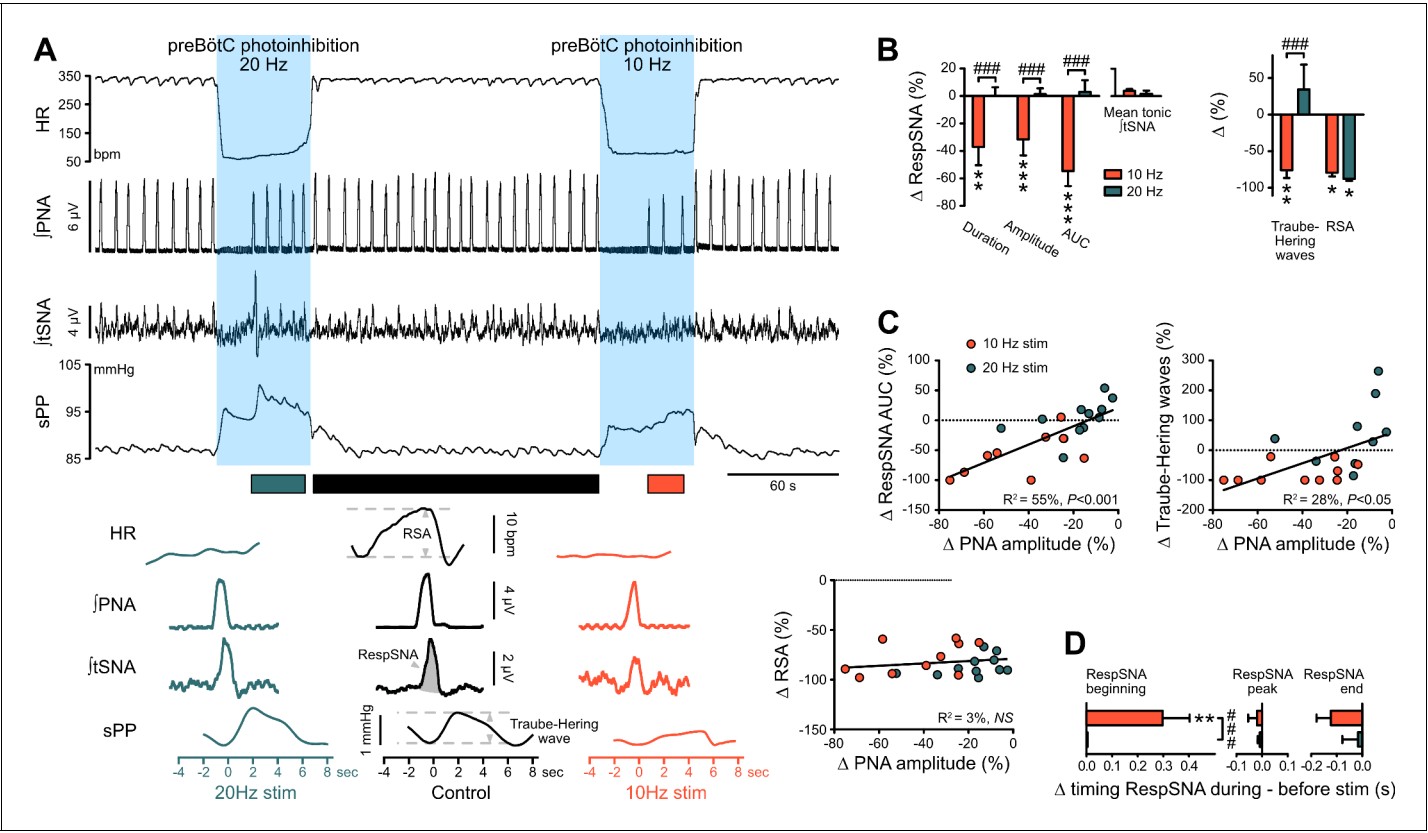

**Figure 6.** The preBötC drives RespSNA, Traube-Hering waves and RSA. (**A**) Traces from one rat and group analysis of the PNA-related oscillations in HR (RSA), tSNA (RespSNA) and systolic PP (Traube-Hering waves). PNA burst-triggered averaging (bottom traces) was performed during control conditions (pre- and post-photoinhibition), and during 10 Hz and 20 Hz photoinhibition when PNA had returned. (**B**) Quantification of the group data showing the effect of preBötC photoinhibition on RespSNA duration, amplitude and area under the curve (AUC), mean tonic tSNA (baseline tSNA outside of RespSNA), Traube-Hering wave amplitude, and RSA. (**C**) Correlation analysis indicates that the PNA amplitude is correlated with RespSNA AUC and Traube-Hering wave amplitude, but not RSA amplitude. The latter indicates that the preBötC generates RSA by a mechanism unrelated to the generation of the PNA inspiratory bursts. The incidence of preBötC photoinhibition on the timing of the onset, peak and end of the RespSNA burst is shown in (**D**). Group data are presented as mean ± SEM; two-way repeated measures ANOVA followed with *post hoc* Holm-Sidak multiple-comparison test performed on raw data, as shown on *Figure 6—figure supplement 1* and associated source data and detailed statistics; **p<0.01, ***p<0.001, photoinhibition vs. control and recovery conditions; ### p<0.001, between photostimulation frequencies.

The online version of this article includes the following source data and figure supplement(s) for figure 6:

**Source data 1.** Source data and statistics for *Figure 6*; *Figure 6—figure supplement 1*.

**Figure supplement 1.** Individual data showing alterations in RespSNA, Traube-Hering waves and RSA during prolonged *Gt*ACR2-mediated preBötC photoinhibition in situ.

## PreBötC photoexcitation decreases the respiratory command amplitude, resulting in decreased RespSNA, Traube-Hering waves and RSA

The same injection protocol described for the *Gt*ACR2 expression was used to produce restricted expression of channelrhodopsin-2 (ChR2) in preBötC bilaterally (*Figure 9A*). Strong ChR2-positive terminals were found in the facial motor nucleus (*Figure 9B*), which is characteristic of neurons in the preBötC region, and not those in the inspiratory rostral ventral respiratory group (*Deschênes et al., 2016*). ChR2-mediated preBötC photoexcitation (n = 8) increased respiratory frequency without altering inspiratory duration, and decreased PNA inspiratory and VNA post-inspiratory amplitudes (*Figure 8A, C and D*), therefore reducing each respiratory cycle intensity. We focused on 20 Hz photoexcitation, as this produced maximal effects across all animals.

PreBötC photoexcitation did not alter overall mean tSNA amplitude (which includes both RespSNA and tonic tSNA) (*Figure 8F*), but disrupted respiratory entrainment of SNA leading to

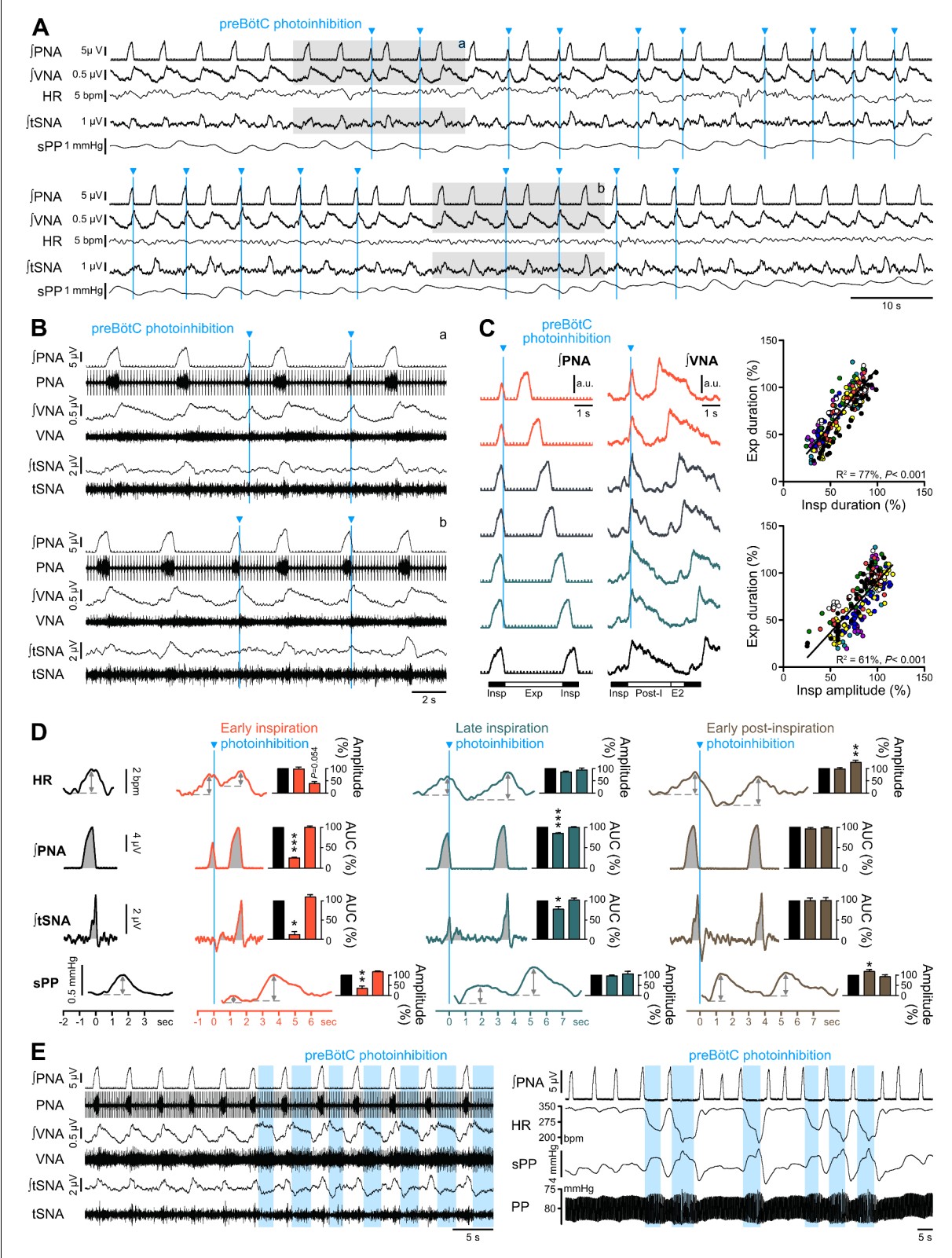

**Figure 7.** Effect of single pulse preBötC photoinhibition on respiration, RespSNA, Traube-Hering waves and RSA. (**A, B**) Traces showing the effect of single light pulse (10 ms) preBötC photoinhibition on phrenic (PNA), vagal (VNA) and thoracic sympathetic (tSNA) nerve activity, HR and systolic perfusion pressure (sPP) at low (**A**) and high (**B**) time resolution in the WHBP. Pulses were applied during early (top traces, (a) and late (bottom traces, (b) inspiration. (**C**) Triggered averages of PNA and VNA showing that single pulse photoinhibition of preBötC delivered during early (orange, 40% and

*Figure 7 continued on next page*

*Figure 7 continued*

50% inspiratory duration), mid- (grey, 60% and 70% inspiratory duration) or late inspiration (green, 80% and 90% inspiratory duration) immediately stopped inspiration (Insp) and triggered post-inspiration (Post-I). For analysis, inspiratory burst duration was normalized to the bottom (black) control trace. The correlograms, showing analysis across multiple respiratory cycles in all animals, with each animal represented by a single color, indicate that the duration of the expiratory phase (Exp, the addition of Post-I and late expiration, (E2) is determined by the duration and amplitude of the preceeding inspiratory period. Data are expressed in % relative to intra-preparation averaged control values, to normalize variability in respiratory phase durations between preparations. (D) PNA burst-triggered averages of HR, PNA, tSNA and sPP were used to derive RSA amplitude, mean PNA AUC, RespSNA AUC and Traube-Hering wave amplitude during control (black) and preBötC photoinhibition in early (orange), late (green) or early post-inspiration (brown). Data are expressed relative to intra-preparation control conditions, and were analyzed for the respiratory cycle during which preBötC photoinhibition occurred and the following, to evaluate direct and indirect photoinhibition effects. (E) Traces showing that when delivered during expiration, preBötC photoinhibition increased VNA for the entire photoinhibition period, but only transiently decreased tSNA. The latter effect was amplified with successive bouts of expiratory preBötC photoinhibition, also inducing increases in RespSNA. PNA bursting always occured following a short (~1 s) post-photoinhibition delay. Group data are presented as mean ± SEM; one-way repeated measures ANOVA followed with *post hoc* Holm-Sidak multiple-comparison test or Friedman repeated measures analysis of variance on ranks followed with pairwise multiple comparison Tukey test performed on raw data, as shown on *Figure 7—figure supplement 1* and associated source data and detailed statistics; *p<0.05, **p<0.01, ***p<0.001, different vs. the two other conditions.

The online version of this article includes the following source data and figure supplement(s) for figure 7:

**Source data 1.** Source data and statistics for *Figure 7*; *Figure 7—figure supplement 1*.

**Figure supplement 1.** Individual data showing respiratory-phase-specific alterations in PNA AUC, RSA, RespSNA and Traube-Hering waves during single-pulse preBötC photoinhibition.

decreased RespSNA amplitude and duration, with shorter inspiratory and post-inspiratory components (*Figure 8I–J*). As a consequence both Traube-Hering wave amplitude (*Figure 8H*) and mean PP (*Figure 8G*) were decreased. Changes in RespSNA magnitude and Traube-Hering wave amplitude induced by preBötC photoexcitation both correlate with changes in PNA inspiratory amplitude (*Figure 8K*). On the other hand, preBötC photoexcitation increased tonic (non-respiratory) tSNA (*Figure 8I*), and laser-triggered averaging of 1 Hz preBötC photoexcitation revealed a principal sympatho-excitatory effect (*Figure 8L–M*).

Photoexcitation of preBötC did not alter mean HR (*Figure 8E*) but decreased RSA amplitude (*Figure 8H*), due to loss of the post-inspiratory/expiratory bradycardic component. Changes in RSA induced by preBötC photoexcitation were independent of changes in PNA inspiratory amplitude (*Figure 8K*). Laser-triggered averaging of VNA during continuous low-frequency (1 Hz) preBötC photoexcitation revealed a small short latency parasympatho-inhibition.

## PreBötC neurons project directly to putative pre-sympathetic vasomotor and parasympathetic neurons

To map axonal projections from preBötC neurons we performed small (5 nl), unilateral virus injections under electrophysiological guidance to induce restricted expression of tdTomato in neurons in the preBötC core (n = 2). Successful targeting of respiratory preBötC neurons was indicated by substantial numbers of transduced fibres crossing the midline and terminal fields within the contralateral preBötC (*Figure 9C*), defining features of functionally identified preBötC neurons (*Bouvier et al., 2010*; *Koizumi et al., 2013*). TdTomato-positive preBötC terminals were found in close apposition to both RVLM catecholaminergic C1 neurons, which include pre-sympathetic vasomotor neurons (*Figure 9D*), and nucleus ambiguus neurons, which include cardiac parasympathetic preganglionic neurons (*Figure 9E*).

## Discussion

Respiratory modulation of sympathetic and parasympathetic activity, leading to Traube-Hering waves and RSA, is a well characterized physiological feature but with little mechanistic understanding. Here, we demonstrate that this respiratory-cardiovascular entrainment is coded at the core of the brainstem respiratory network. In addition to its role in inspiratory rhythm generation, the preBötC provides inspiratory-locked excitatory drive to sympathetic activity and subsequent generation of Traube-Hering waves, and inhibitory drive to parasympathetic outputs, to generate RSA. Photoinhibition of preBötC exerts profound effects on HR and BP in both anesthetized in vivo and in situ rat models, highlighting the importance of this ongoing activity under basal conditions. These data

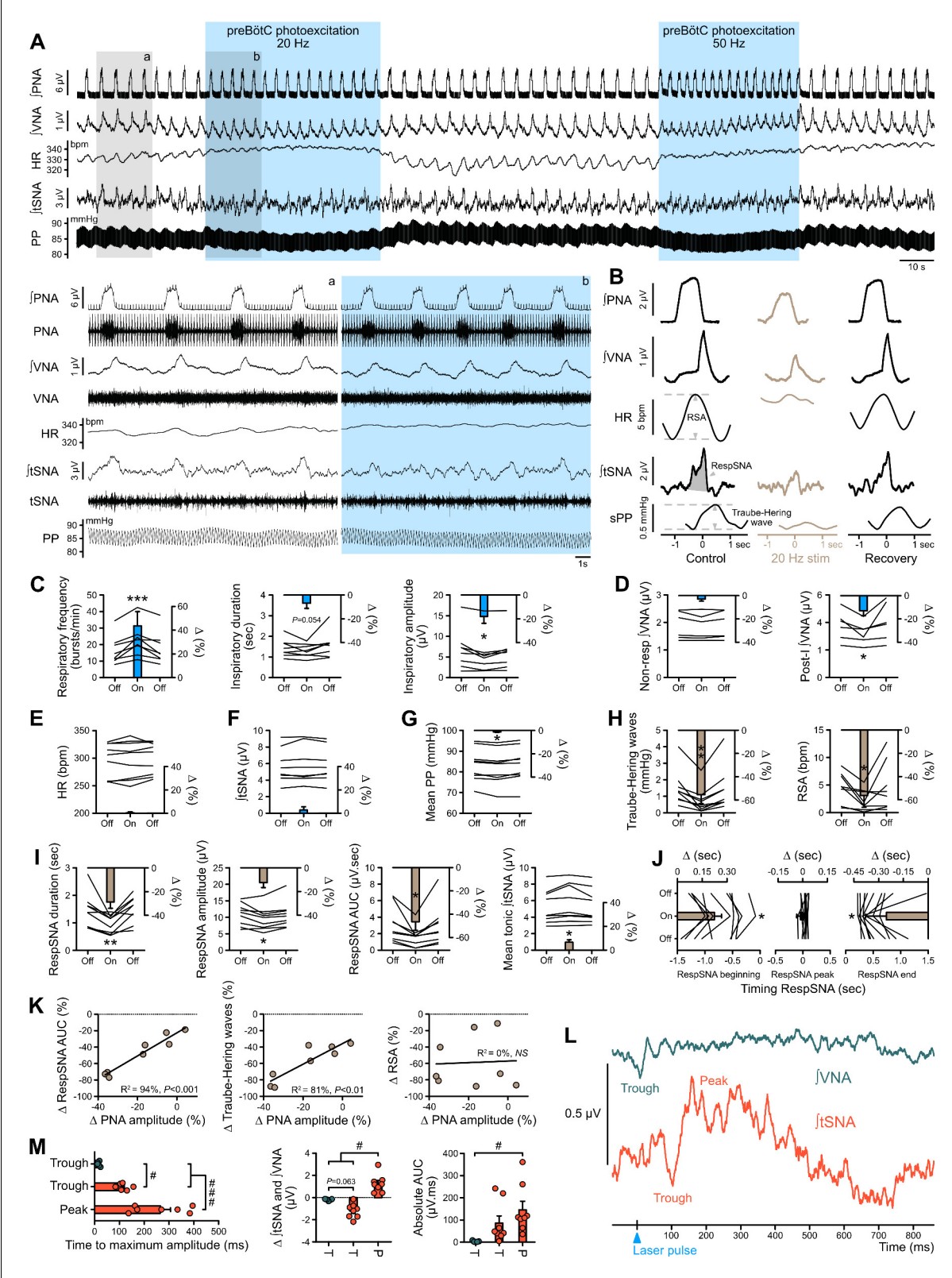

**Figure 8.** Photoexcitation of preBötC decreases amplitude of the respiratory command, RespSNA, Traube-Hering waves and RSA. Traces of activity recorded in the WHBP at low (**A**) and higher (lower traces from the shaded areas) time resolution showing the effect of bilateral photoexcitation of the preBötC with ChR2. (**B**) PNA burst-triggered averaging was used to quantify the effect on RespSNA AUC, Traube-Hering wave amplitude and RSA amplitude. (**C–J**) Individual and group data (n = 8) quantifying the effect of preBötC photoexcitation (20 Hz) on: (**C**) respiratory parameters; (**D**) VNA; (**E**)

*Figure 8 continued on next page*

*Figure 8 continued*

HR; (F) tSNA; (G) mean PP; (H) Traube-Hering wave and RSA; and (I) RespSNA. (J) Further analysis shows the decrease in duration of RespSNA was due to both shorter inspiratory and post-inspiratory components. (K) Correlograms showing that RespSNA AUC and Traube-Hering wave amplitude were both correlated with PNA amplitude, whilst RSA was not. (L) Traces from one animal showing laser-triggered averaging of integrated VNA and tSNA in response to 1 Hz preBötC photoexcitation. (M) Quantification of this response in all animals showing photoexcitation induces inverse responses to that of photoinhibition, and of smaller amplitude. Group data are presented as mean ± SEM; one-way repeated measures ANOVA followed with *post hoc* Holm-Sidak multiple-comparison test or Friedman repeated measures analysis of variance on ranks followed with pairwise multiple comparison Tukey test performed on raw data, as shown on the associated source data and detailed statistics; *p<0.05, **p<0.01, ***p<0.001, photoinhibition vs. control and recovery conditions; # p<0.05, ## p<0.01, between respective effects.

The online version of this article includes the following source data for figure 8:

**Source data 1.** Source data and statistics for *Figure 8*.

reinforce other recent observations (*Deschênes et al., 2016*; *Moore et al., 2013*) supporting the view that oscillatory activity arising from the preBötC provides widespread influences extending well past circuits related to breathing.

## Targeting the preBötC

The preBötC was defined as a group of excitatory inspiratory neurons with intrinsic rhythmogenic properties, located in the ventral respiratory column (*Peña et al., 2004*; *Smith et al., 1991*). It remains classified as a region with relatively ambiguous anatomic boundaries that contains a hetero-geneous cell population composed of rhythmogenic, mostly inspiratory, excitatory and inhibitory neurons (*Baertsch et al., 2019*; *Baertsch et al., 2018*; *Koizumi et al., 2013*; *Kuwana et al., 2006*; *Morgado-Valle et al., 2010*; *Sherman et al., 2015*). No single and exclusive marker of the preBötC has been found so far. We studied neurons in the preBötC using the pan-neuronal CBA promoter that targets neurons irrespective of phenotype. Our anatomical validation of correct transduction (*Figures 1* and *9*) is supported functionally, as photoinhibition caused immediate cessation of inspi-ratory activity and a prolonged apnea both in vivo and in situ (*Figures 2* and *3*), whereas photoexci-tation increased respiratory frequency while decreasing inspiratory amplitude (*Figure 8*). These effects on breathing are consistent with previous optogenetic perturbations of the preBötC (*Alsahafi et al., 2015*; *Cui et al., 2016*; *Koizumi et al., 2016*; *Vann et al., 2018*). Adjacent to the preBötC are caudal ventrolateral medulla (CVLM) GABAergic neurons, which form part of the sym-pathetic vasomotor circuit and inhibit rostral ventrolateral medulla pre-sympathetic neurons (*Guye-net, 2006*), as well as some cardiac parasympathetic preganglionic neurons of the nucleus ambiguus (*Dergacheva et al., 2010*; *Gourine et al., 2016*). Some of these neurons could have been trans-duced with our viral approach; however, our physiological observations do not support this being a substantial influence, as *Gt*ACR2-mediated photoinhibition caused inverse effects to those expected if these neurons were transduced (which would be sympathoexcitation/increase in blood pressure and parasympatho-inhibition/tachycardia) (*Figures 3* and *5*).

When inspiration resumed during photoinhibition, we observed a crenelated activity pattern in inspiratory intC EMG and PNA bursts (*Figures 2* and *3*). This is likely to result from photoinhibition of some rostral ventral respiratory group phrenic premotor neurons, which intermingle with the cau-dal pole of the preBötC. Importantly, these neurons are not thought to project to respiratory/cardio-vascular neurons other than phrenic motoneurons and lateral reticular nucleus (*Wu et al., 2017*), so they are unlikely to mediate the cardiac parasympathetic and vasomotor sympathetic effects found in this study.

## Caveats related to optically-activated chloride channels

To interrogate interactions between preBötC respiratory circuits and autonomic outputs we used the high temporal resolution of the natural light-activated anion channel *Gt*ACR2, which induces large chloride photocurrents with faster kinetics than other inhibitory opsins (*Govorunova et al., 2015*; *Figure 2*). Interpretation of our results requires careful consideration, as the chloride electro-chemical gradient is heterogeneous among subcellular compartments, and axonal activation of chlo-ride channels can lead to terminal neurotransmitter release (*Messier et al., 2018*). This phenomenon is unlikely to explain the current results, which are consistent with neural inhibition and were opposed by optogenetic excitation, highlighted by the strong, perfectly inverse sympathetic (tSNA)

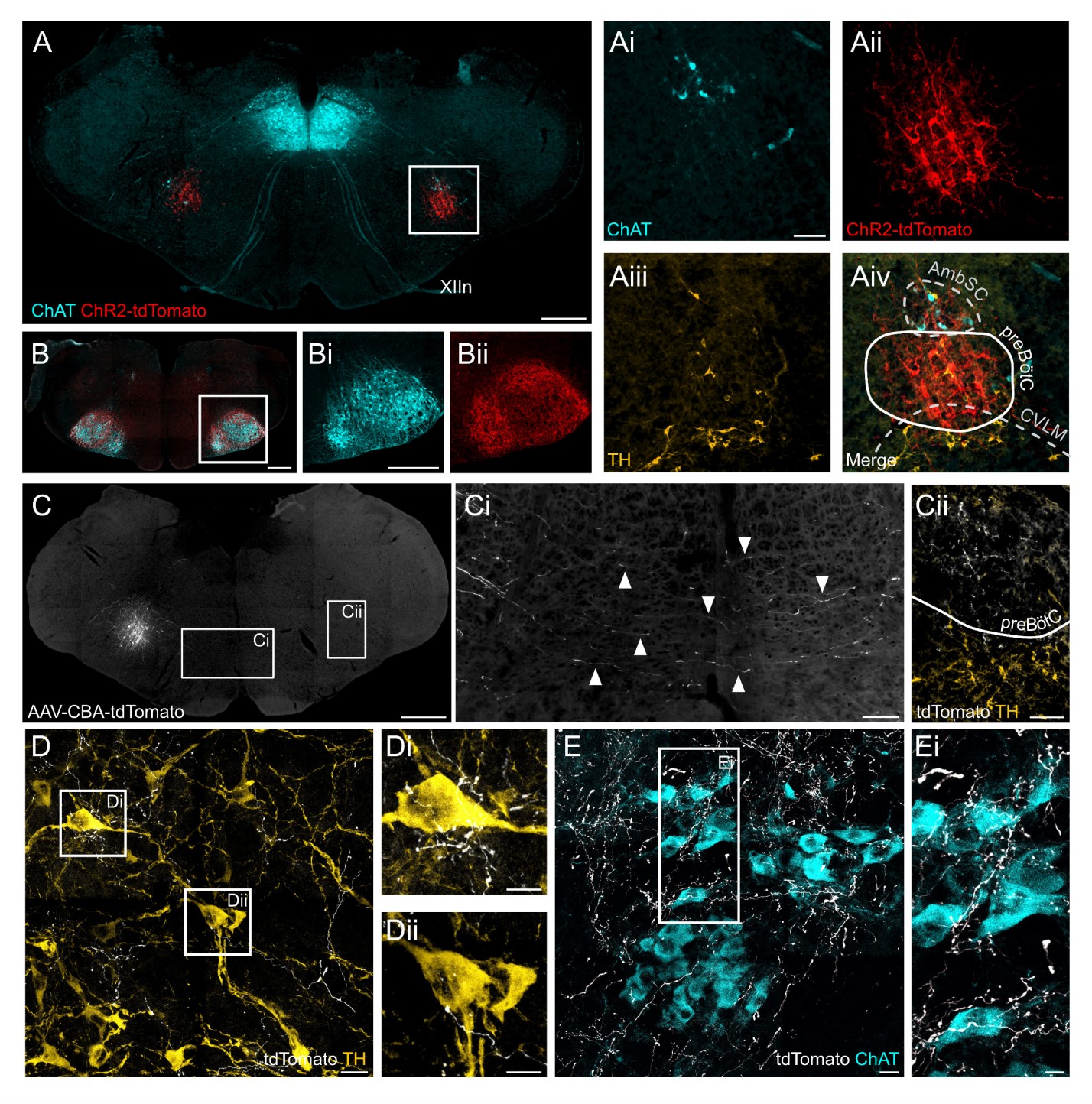

**Figure 9.** ChR2 expression in the preBötC, and direct projections to adrenergic C1 and nucleus ambiguus neurons. (**A**) Coronal section showing immunoreactivity for tdTomato, fused to ChR2, choline acetyltransferase (ChAT) and tyrosine hydroxylase (TH) at low and high magnification (white box in A, shown as individual colour channels in Ai-iv). The ChR2 expression profile is restricted to the preBötC, ventral to the semicompact formation of the nucleus ambiguus (AmbSC), and dorsal to the caudal ventrolateral medulla oblongata (CVLM). (**B**) ChR2-positive terminals were found in the facial nucleus, a characteristic feature of preBötC projections (*Deschênes et al., 2016*). (**C**) Small volume (5 nl) unilateral viral injections (n = 2) restricted tdTomato expression to core preBötC neurons, with substantial numbers of tdTomato-positive fibers crossing the midline (Ci) to the contralateral preBötC (Cii). (**D**) TdTomato-positive preBötC terminals were found in close apposition to TH-positive C1 neurons in the rostral ventrolateral medulla (Di and Dii), and (**E**) ChAT-positive neurons in the compact formation of the nucleus ambiguus (AmbC) (Ei). Scale bars are 500 µm (**A, B, Bi, C**), 100 µm (Ai, Ci, Cii), 20 µm (**D, E**) or 10 µm (Di, Dii, Ei).

and parasympathetic (VNA) effects observed with single light pulse ChR2-mediated photoexcitation (*Figures 5* and *8*). We conclude that in this study and with this cell group, *Gt*ACR2 reliably inhibited neuronal activity.

## The preBötC provides powerful, ongoing inhibition of cardiac parasympathetic activity

Cardiac parasympathetic preganglionic neurons are located in the nucleus ambiguus and the dorsal motor nucleus of the vagus nerve (*Gourine et al., 2016*). Among these, only nucleus ambiguus neurons show respiratory-related phasic patterns of activity, with an inhibition of their activity during inspiration, an excitation during post-inspiration and a weak inhibition during late expiration (E2) (*Gilbey et al., 1984*; *Neff et al., 2003*). Respiratory inputs into nucleus ambiguus cardiac preganglionic neurons could be widely distributed, although the strongest evidence indicates they come from the pontine Kölliker-Fuse nucleus for the post-inspiratory excitation (*Farmer et al., 2016*), and from GABAergic and glycinergic neurons that could be located in the preBötC for the inspiratory inhibition (*Frank et al., 2009*; *Neff et al., 2003*).

Single light pulse preBötC photoinhibition induced intense vagal activation (*Figure 7*), whilst prolonged photoinhibition, both in vivo and in situ, caused profound bradycardia, with up to a 70% decrease in HR (*Figures 2* and *3*), which was blocked by of the parasympatholytic agent atropine (*Figure 4*). The simplest pathway that could explain this result is a direct inhibitory connection between preBötC and cardiac vagal motoneurons, a scheme supported by our tracing studies (*Figure 9*) and the short latencies of vagal responses to optical excitation and inhibition (*Figures 5* and *8*). The preBötC contains both GABAergic and glycinergic neurons (*Koizumi et al., 2013*; *Kuwana et al., 2006*; *Morgado-Valle et al., 2010*) that display four activity patterns: tonic, inspiratory, pre-inspiratory and expiratory (*Baertsch et al., 2018*; *Morgado-Valle et al., 2010*; *Sherman et al., 2015*). It is likely that most of these functional subgroups are involved in cardiac parasympatho-modulation.

Bradycardia occurs independent of when in the respiratory cycle photoinhibition commences, showing that tonic neurons must be involved. Most interestingly, RSA, which is mediated by a progressive decrease of vagal parasympathetic drive during expiration that peaks during inspiration (*Bouairi et al., 2004*; *Dergacheva et al., 2010*; *Gilbey et al., 1984*; *Neff et al., 2003*), was nearly abolished by preBötC photoinhibition, even after the re-commencement of PNA during prolonged photoinhibition. RSA was also abolished with prolonged preBötC photoexcitation, while there was a tendency for an increase in HR (*Figure 8*). This shows that prolonged photostimuli of the preBötC, that are likely to suppress phasic activities in targeted neurons and render them either inactive (photoinhibition) or tonically active (photoexcitation), are able to suppress RSA. This could be explained by suppression of the phasic activity of inhibitory preBötC neurons, composed of a small (~10%) proportion of inhibitory expiratory neurons (*Baertsch et al., 2018*; *Carroll et al., 2013*), and a larger proportion of inhibitory pre-inspiratory and inspiratory neurons. As a result, phasic late expiratory and inspiratory inhibition of cardiac vagal parasympathetic activity would be suppressed, therefore suppressing RSA. The preBötC is therefore critically involved in tonic regulation of HR as well as the generation of RSA, through a combination of phasic inhibitory neurons.

The decrease in HR just after inspiration, during the post-inspiratory phase, likely involves an excitatory drive to cardiac parasympathetic preganglionic neurons external to the preBötC, most likely coming from the pontine Kölliker-Fuse nucleus (*Farmer et al., 2016*; *Gilbey et al., 1984*). Still, preBötC neurons exert an indirect influence on this, as a single photoinhibitory light pulse during inspiration modulated both the inspiratory and post-inspiratory phases, as well as amplitude of the following cycle of RSA, by reducing the post-inspiratory decrease in HR (*Figure 7*).

## The preBötC provides excitatory drive to vasomotor sympathetic activity

Respiratory modulation of neurons involved in generating vasomotor sympathetic activity is found from spinal sympathetic preganglionic neurons of the intermediolateral cell column (*Zhou and Gilbey, 1992*) to medullary bulbospinal pre-sympathetic neurons of the RVLM (*McAllen, 1987*; *Moraes et al., 2013*) and caudal raphe (*Gilbey et al., 1995*), as well as in baroactivated CVLM neurons (*Mandel and Schreihofer, 2006*). Neurons in each of these regions display three main types of

respiratory modulation, inspiratory activated, inspiratory inhibited, and post-inspiratory activated, which could be acquired mono-synaptically or poly-synaptically from respiratory neurons.

Prolonged preBötC photoinhibition reduced SNA and PP, but only for the apneic period (*Figure 3*). Single-pulse preBötC photoinhibition revealed a fast inhibition of SNA, peaking ~140 ms after the light pulse (*Figure 5*). This effect on SNA is delayed compared to the latencies of responses recorded in VNA, and cannot be attributed to the relatively slow conduction velocity of sympathetic pathways, as SNA responses to photostimulation of sympathetic premotor neurons are also lower than those seen here (*Abbott et al., 2009*; *Menuet et al., 2014*). These observations are consistent with a polysynaptic pathway prior to the recorded SNA, which could potentially include recruitment of neurons in the intermediate reticular nucleus (*Toor et al., 2019*). Also, inhibitory, GABAergic CVLM neurons are respiratory modulated (*Mandel and Schreihofer, 2006*) and could mediate inspiratory modulation of RVLM pre-sympathetic neurons arising from inhibitory or excitatory inspiratory preBötC neurons. Microinjection of the GABA$_A$ receptor agonist muscimol into the CVLM/preBötC abolishes RespSNA and respiratory modulation of RVLM neurons, but it is not clear whether this is a direct effect to CVLM inhibition, or due to the associated total loss of respiratory drive and consequent network effects (*Koshiya and Guyenet, 1996*). Microinjections of GABA$_A$ receptor antagonists in the RVLM enhance RespSNA (*Guyenet et al., 1990*; *Menuet et al., 2017*), which, together with our single-pulse preBötC photoinhibition experiments, favor the hypothesis that any respiratory phasic input from the CVLM is likely to inhibit presympathetic RVLM neurons.

Direct recordings show that inspiratory modulation of RVLM C1 neurons is mediated by excitatory post-synaptic potentials (*Moraes et al., 2013*), and we found evidence that neurons in the preBötC, albeit not definitively demonstrated to be inspiratory-modulated neurons, send direct projections onto C1 neurons (*Figure 9*). This confirms our previous data showing retrograde trans-synaptic labeling of preBötC NK1R neurons from pre-sympathetic C1 neurons (*Dempsey et al., 2017*; *Menuet et al., 2017*). Upon resumption of PNA, but during continued photoinhibition, the amplitude of RespSNA and Traube-Hering waves were decreased, strongly correlating with PNA amplitude (*Figure 6*). Single-pulse preBötC photoinhibition during inspiration decreased the concomitant RespSNA burst and subsequent Traube-Hering wave (*Figure 7*). During preBötC photoexcitation, inspiratory amplitude was decreased and this also correlated with decreased RespSNA and Traube-Hering waves amplitudes (*Figure 8*). We therefore conclude the preBötC provides ongoing excitatory drive to vasomotor sympathetic activity, most likely via pre-sympathetic C1 neurons whose activity correlates with inspiration (*Menuet et al., 2017*; *Moraes et al., 2013*), and drives RespSNA and Traube-Hering waves (*Marina et al., 2011*; *Menuet et al., 2017*; *Moraes et al., 2017*).

Single-pulse preBötC photoinhibition had no effect on RespSNA when applied during post-inspiration (*Figure 7*). Part of the post-inspiratory component of RespSNA originates from post-inspiratory neurons in the intermediate reticular nucleus (*Toor et al., 2019*), and similarly to RSA, the dorsolateral pons could also be involved (*Baekey et al., 2008*; *Dick et al., 2009*). Indeed, pontomedullary transections strongly decreased RespSNA and Traube-Hering waves. Yet, this also caused a switch from eupnea to an apneustic breathing pattern, with prolonged and square shaped phrenic nerve activity. This highlights the difficulty to interrogate functional circuitry entrained by respiratory neurons, as alteration of the activity of the respiratory network will create rebound network effects and alterations of breathing, making it difficult to disentangle the direct versus indirect mechanisms inlvolving the neurons studied. Ideally, future experiments will enable modulation of individual synapses between respiratory neurons and neurons regulating sympathetic (or parasympathetic) activity.

## Different preBötC respiratory neurons affect SNA compared to cardiac vagal activity

The current view is that excitatory preBötC neurons generate the inspiratory rhythm (*Bouvier et al., 2010*; *Vann et al., 2018*; *Wang et al., 2014*), while inhibitory preBötC neurons shape its pattern and frequency (*Baertsch et al., 2018*; *Cregg et al., 2017*; *Sherman et al., 2015*). Here, prolonged preBötC photoinhibition induced apnea but not for the entire photoinhibition period. Previous investigators have reported prolonged apnea upon preBötC inhibition induced by pharmacogenetic (*Tan et al., 2008*) or (brief) optogenetic neural inhibition (*Vann et al., 2018*). Like the inspiratory activity, tSNA also recovered respiratory modulation when inspiration resumed. By stark contrast, vagal activation and bradycardia remained affected for the entire period of photoinhibition. This

observation provides strong support for affecting two different cell populations, an excitatory one involved in inspiratory rhythmogenesis and SNA modulation, and an inhibitory one involved in inspiratory patterning and cardiac vagal modulation. The most likely explanation for the return of respiratory activity during continued photoinhibition is that rhythmogenic, excitatory inspiratory neurons of the preBötC escaped GtACR2-mediated photoinhibition. The mechanism responsible for the rhythmogenic activity of preBötC is still debated, but involves excitatory pre-inspiratory and inspiratory neurons with cellular and/or micro-network electrophysiological properties particularly prone to burst generation, which could confer particular propensity to escape prolonged inhibition. If this is the mechanism, it appears to be a property of the excitatory neurons.

### Respiratory cardiovascular interactions in the clinical setting

Assessment of the short-term variability of HR, and to a lesser extent BP, is common in clinical evaluation of cardiovascular health. The main spectral density found in both HR and BP variability is in phase with the respiratory frequency, in the high-frequency band, and is actually a quantification of RSA and Traube-Hering waves, respectively. These measurements provide an invaluable means for non-invasive assessment of parasympathetic and sympathetic drives to the cardiovascular system and have attracted general community interested due to the connection between HR variability and 'healthiness' scores. However, views on the analysis and interpretation of these data vary substantially (*Hayano and Yuda, 2019*). At the core of uncertainty regarding the value of HR and BP variability lies a lack of understanding of the mechanisms driving RSA and Traube-Hering waves.

Regardless of how it is measured, substantial evidence links altered respiratory modulation of autonomic activity with cardiovascular diseases. Exaggerated inspiratory modulation of sympathetic activity leads to increased Traube-Hering waves and drives the development of hypertension (*Menuet et al., 2017*). This study identifies a key source of this excitatory inspiratory modulation, and while modifying preBötC activity is not conceivable given its critical role, targeted therapeutics that only impact the preBötC to C1 neuron connection could be developed to prevent hypertension development. A common feature of many cardiovascular diseases, and prognostic indicator of poor survival, is reduced RSA, often in concert with increased Traube-Hering waves (*Palatini and Julius, 2009*; *Task Force of the European Society of Cardiology and the North American Society of Pacing and Electrophysiology, 1996*). Our study also identifies the preBötC as the likely source of this dual alteration of cardiac parasympathetic and vasomotor sympathetic respiratory drives.

### Conclusion

Whilst initially characterized as a key oscillator for driving inspiratory activity, the preBötC has already been shown to have more widespread functions, acting as a master oscillator driving other activities, such as sniffing and whisking (*Deschênes et al., 2016*; *Moore et al., 2013*). Redundancy is usually a characteristic feature of neuronal networks, as it provides robustness to perturbations (*Li et al., 2016*). It is therefore surprising to find that the preBötC is so critical for respiratory, cardiovascular and other oscillations. Our study provides evidence for another layer of influence for the preBötC as a hub oscillator that synchronizes cardio-respiratory function. This coordination is a fundamental physiological phenomenon observed across many different phyla that is likely to be of key importance for optimal perfusion, and function, of tissues.

## Materials and methods

**Key resources table**

| Reagent type (species) or resource | Designation | Source or reference | Identifiers | Additional information |
|---|---|---|---|---|
| Strain, strain background (male Sprague Dawley rats) | WT Sprague Dawley rats | Biomedical Science Animal Facility of the University of Melbourne | | |

*Continued on next page*

*Continued*

| Reagent type (species) or resource | Designation | Source or reference | Identifiers | Additional information |
|---|---|---|---|---|
| Recombinant DNA reagent | pFUGW-hGtACR2-EYFP | Addgene, gift from John Spudich | plasmid # 67877; RRID:Addgene_67877 | http://n2t.net/addgene:67877 |
| Recombinant DNA reagent | pAAV-CAG-hChR2-H134R-tdTomato | Addgene, gift from Karel Svoboda | plasmid #28017; RRID:Addgene_28017 | http://n2t.net/addgene:28017 |
| Recombinant DNA reagent | pAM.DCA.spe.tdTomato | Gift from Verena Wimmer | | |
| Strain, strain background (lentivirus) | Lv-CBA-GtACR2-YFP-WPRE | In house cloning and virus production | | Titre $6.95 \times 10^9$ IU/ml |
| Strain, strain background (AAV) | AAV-CBA-GtACR2-YFP-WPRE | In house cloning and virus production | | Titre $9.65 \times 10^{11}$ VP/ml |
| Strain, strain background (AAV) | AAV-CAG-hChR2-H134R-tdTomato | In house virus production | | Titre $2.23 \times 10^{10}$ VP/ml |
| Strain, strain background (AAV) | AAV-CBA-tdTomato | In house virus production | | Titre $6.32 \times 10^{12}$ VP/ml |
| Antibody | Rabbit polyclonal anti-TH | Merck-Millipore Bioscience Research Reagents | AB152 | 1:5000 |
| Antibody | Chicken polyclonal anti-GFP | Abcam | AB13970 | 1:5000 |
| Antibody | Mouse monoclonal anti-parvalbumin | Merck-Millipore Bioscience Research Reagents | MAB1572 | 1:10,000 |
| Antibody | Rabbit polyclonal anti-NK1R | Merck-Sigma-Aldrich | S8305 | 1:5000 |
| Antibody | Rabbit polyclonal anti-DsRed | Takara Bio | 632496 | 1:5000 |
| Antibody | Goat polyclonal anti-ChAT | Merck-Millipore Bioscience Research Reagents | AB144P | 1:1000 |
| Antibody | Rabbit polyclonal anti-GFP | Life Technologies | A-6455 | 1:500 |
| Antibody | Cy3-conjugated donkey polyclonal anti-rabbit | Jackson ImmunoResearch Laboratories | 711-165-152 | 1:500 |
| Antibody | Cy3-conjugated donkey polyclonal anti-mouse | Jackson ImmunoResearch Laboratories | 715-165-151 | 1:500 |
| Antibody | AlexaFluor-488 donkey polyclonal anti-rabbit | Jackson ImmunoResearch Laboratories | 711-545-152 | 1:500 |

*Continued on next page*

*Continued*

| Reagent type (species) or resource | Designation | Source or reference | Identifiers | Additional information |
|---|---|---|---|---|
| Antibody | AlexaFluor-488 donkey polyclonal anti-mouse | Jackson ImmunoResearch Laboratories | 715-545-151 | 1:500 |
| Antibody | AlexaFluor-488 donkey polyclonal anti-chicken | Jackson ImmunoResearch Laboratories | 703-545-155 | 1:500 |
| Antibody | AlexaFluor-488 donkey polyclonal anti-rabbit | Life Technologies | A21206 | 1:500 |
| Sequence-based reagent | PCR primers for generation of cRNA probe for glycine transporter 2 (GlyT2) | | | SP6 forward primer: GGATCCATTTAGGT GACACTATAGAAG aagcgtcttgcccactagaa T7 reverse primer: GAATTCTAATACGA CTCACTATAGGGA GAagcctgagcttgcttttcag |
| Sequence-based reagent | PCR primers for generation of cRNA probe for glutamic acid decarboxylase 67 (GAD67) | | | SP6 forward primer: GGATCCATTTAGGTGAC ACTATAGAAGttatgtc aatgcaaccgc T7 reverse primer: GAATTCTAATACGACTCAC TATAGGGAGAcccaac ctctctatttcctcC |
| Sequence-based reagent | PCR primers for generation of cRNA probe for vesicular glutamate transporter 2 (VGluT2) | | | SP6 forward primer: GGATCCATTTAGGT GACACTATAGAAGt caatgaaatccaacgtcca T7 reverse primer: GAATTCTAATACGA CTCACTATAGGGA GAcaagagcacag gacaccaaa |
| Chemical compound, drug | Meloxicam | Boehringer Ingelheim | | 1 mg/kg, s.c. |
| Chemical compound, drug | Isoflurane | Rhodia Australia Pty. Ltd., | | 5% induction, 3% maintenance |
| Chemical compound, drug | Ketamine | Lyppard | | 60 mg/kg, i.m. |
| Chemical compound, drug | Medetomidine | Pfizer Animal Health | | 250 μg/kg, i.m. |
| Chemical compound, drug | Atipamazole | Pfizer Animal Health | | 1 mg/kg, i.m. |
| Chemical compound, drug | Urethane | SigmaAldrich | | 1.2 mg/kg i.v. |
| Chemical compound, drug | Vasopressin acetate | Sigma-Aldrich | V9879 | 0.5 nM |

*Continued on next page*

*Continued*

| Reagent type (species) or resource | Designation | Source or reference | Identifiers | Additional information |
|---|---|---|---|---|
| Chemical compound, drug | Vecuronium bromide | Organon Teknika | | 2–4 µg.ml$^{-1}$ |
| Chemical compound, drug | Atropine | Sigma Aldrich | A0257-25G | 0.2 µM |
| Software, algorithm | Spike2 | Cambridge Electrical Design | | |
| Software, algorithm | pClamp 10.3 | Molecular Devices | | |
| Software, algorithm | ZEN 2.6 | Carl-Zeiss | | |
| Software, algorithm | Affinity Designer | Serif Ltd. | | |
| Software, algorithm | Image J | NIH | | |
| Software, algorithm | SigmaPlot v12 | Systat Software Inc | | |
| Software, algorithm | Prism v2.0 | GraphPad | | |

## Animal experiments

Experiments were conducted in accordance with the Australian National Health and Medical Research Council 'Code of Practice for the Care and Use of Animals for Scientific Purposes' and were approved by the University of Melbourne Animal Research Ethics and Biosafety Committees (ethics ID #1413273, #1614009, #1814599 and Florey 16-040). All experiments were performed on male Sprague Dawley (SD) rats. Animals were housed with a 12-h light-dark cycle (06H30 to 18H30), at a constant temperature (22 ± 1°C) with ad libitum access to standard rat chow and water.

## Viruses

To perform optogenetic inhibition or excitation of preBötC neurons, we used viral-mediated transgenesis to drive the expression of two opsins, the guillardia theta anion channelrhodopsin 2 (*Gt*ACR2) and the humanized ChR2 H134R, fused to the enhanced yellow fluorescent protein (YFP) and tdTomato reporter genes, respectively. To trace preBötC terminals, a virus driving only tdTomato expression was used. We used the pan-neuronal promoter chicken ß-actin (CBA), either alone, or as part of the larger synthetic promoter CAG. Indeed, CAG induces the same expression profile as CBA, since CBA is the promoter component (A) of CAG (C being the cytomegalovirus early enhancer element, G being the splice acceptor of the rabbit ß-globulin gene). The viral vectors used were: Lv-CBA-*Gt*ACR2-YFP-WPRE (titre 6.95x10$^9$ IU/ml), AAV-CBA-*Gt*ACR2-YFP-WPRE (titre 9.65x10$^{11}$ VP/ml), AAV-CAG-hChR2-H134R-tdTomato (titre 2.23x10$^{10}$ VP/ml), AAV-CBA-tdTomato (6.32x10$^{12}$ VP/ml). The *Gt*ACR2 constructs were cloned in-house from the pFUGW-hGtACR2-EYFP plasmid, which was a gift from John Spudich (Addgene plasmid # 67877; http://n2t.net/addgene:67877; RRID:Addgene_67877) (*Govorunova et al., 2015*), by replacing the UbiquitinC promoter with CBA. The ChR2-expressing virus was made from the pAAV-CAG-hChR2-H134R-tdTomato plasmid, which was a gift from Karel Svoboda (Addgene plasmid #28017; http://n2t.net/addgene:28017; RRID:Addgene_28017) (*Mao et al., 2011*). The tdTomato-expressing virus was made from pAM. DCA.spe.tdTomato plasmid.

## Brainstem virus microinjections

Thirty to sixty minutes prior to surgery, 21-day-old (P21) rats were injected with a non-steroidal, anti-inflammatory drug (meloxicam, 1mg/kg, s.c., Metacam, Boehringer Ingelheim, Sydney, Australia)

and then lightly anesthetized by inhalation of isoflurane in an induction box (Rhodia Australia Pty. Ltd., Notting Hill, Australia) prior to intramuscular injection of a mixture of ketamine (60mg/kg, i.m.; Lyppard, Dingley, Australia) and medetomidine (250 µg/kg, i.m.; Pfizer Animal Health, West Ryde, Australia). Surgery was initiated once a deep surgical level of anesthesia was obtained, as evidenced by loss of the pedal withdrawal and corneal reflexes. Body temperature was maintained at 37.5°C with a heat pad (TC-1000 Temperature controller, CWE Inc, USA) that was covered with an auto-claved absorbent pad. Rats were then placed in a stereotaxic frame with the nose ventro-flexed (40°). A midline incision was made over the occipital bone, portions of the bone overlying the cerebellum were removed with a dental drill, and the atlanto-occipital membrane was opened to expose the dorsal brainstem. Glass micropipettes (~30 µm tip diameter) were back-filled with injectate and connected via a silver wire to an electrophysiology amplifier (NeuroLog system, Digitimer Ltd., Welwyn Garden City, UK) to enable recording of neuronal activity. The pipette (20° angled in the rostro-caudal axis, tip pointing forward) was descended into the brainstem 1.5 mm lateral to the midline and extracellular recordings of multiunit activity used to functionally map the preBötC, defined as the most rostral point at which vigorous inspiratory-locked activity was isolated (typically 0.2 ± 0.3 mm rostral, 1.4 ± 0.3 mm ventral to the *calamus scriptorius*). Bilateral injections were performed for *Gt*ACR2 and ChR2 expressions, unilateral injection for tdTomato expression. For in vitro slice electrophysiology, *Gt*ACR2 expression was induced in the solitary tract nucleus (NTS). NTS injection coordinates, relative to *calamus scriptorius* for antero-posterior and medio-lateral coordinates, with all injections performed 0.4mm ventral to the dorsal surface of the brainstem at each injection coordinate: 0.4mm and 0.1mm caudal (midline); 0.1mm rostral, 0.25mm lateral (both sides); 0.4mm rostral, 0.4mm lateral (both sides). A picospritzer (World Precision Instruments, Sarasota, USA) was used to microinject viral vectors driving GtACR2-YFP (50 nl per injection site over 5 min), ChR2-tdTomato (100 nl per injection site over 1 min) or tdTomato alone (5 nl per injection site over 1 min). Pipettes were left in place for 5 min after injection, the pipette withdrawn, and wounds closed with sterile sutures, and anesthesia reversed with atipamazole (1 mg/kg, i.m., Antisedan, Pfizer Animal Health, West Ryde, Australia). Physiological experiments (working heart and brainstem preparation (WHBP), n = 12 *Gt*ACR2 and n = 8 ChR2; in vivo electrophysiology, n = 5 Lv-*Gt*ACR2 and n=3 AAV-*Gt*ACR2; and in vitro whole cell recordings, n = 7) were performed 10 days after viral injections, while animals used for anterograde tracing (n = 2) were sacrificed 20 days after vector injection.

## In vivo electrophysiology

Lv-GtACR2 injected rats (n = 5), *Figure 2A-D*: 30 to 60 min prior to surgery, rats were injected with a non-steroidal, anti-inflammatory drug (meloxicam, 1mg/kg, s.c., Metacam, Boehringer Ingelheim, Sydney, Australia) and then lightly anesthetized by inhalation of isoflurane in an induction box (Rhodia Australia Pty. Ltd., Notting Hill, Australia) prior to intramuscular injection of a mixture of ketamine (60mg/kg, i.m.; Lyppard, Dingley, Australia) and medetomidine (250 µg/kg, i.m.; Pfizer Animal Health, West Ryde, Australia). Surgery was initiated once a deep surgical level of anesthesia was obtained, as evidenced by loss of the pedal withdrawal and corneal reflexes. Body temperature was maintained at 37.5°C with a heat pad (TC-1000 Temperature controller, CWE Inc, USA) that was covered with an autoclaved absorbent pad. Rats were placed in a stereotaxic frame with the nose slightly dorso-flexed (mouth bar set at 0). A small incision was made over the dorso-lateral thorax and bipolar stainless steel hook electrodes were placed in consecutive intercostal muscles with a ground electrode positioned under the skin. Recordings of inspiratory intercostal activity were amplified, filtered (50–1500 Hz band-pass, NeuroLog system, Digitimer Ltd., Welwyn Garden City, UK), sampled (5 kHz) and digitized (Power 1401, Cambridge Electrical Design, Cambridge, UK), and rectified and integrated using Spike2 software (Cambridge Electrical Design, Cambridge, UK). Heart rate (HR) was derived from the same biopotential using a window discriminator to trigger from the R-wave of the electrocardiogram.

AAV-GtACR2 injected rats (n = 3), *Figure 2—figure supplement 1*: 30 to 60 min prior to surgery, rats were injected with a non-steroidal, anti-inflammatory drug (meloxicam, 1mg/kg, s.c., Metacam, Boehringer Ingelheim, Sydney, Australia) and then lightly anesthetized by inhalation of isoflurane in an induction box (Rhodia Australia Pty. Ltd., Notting Hill, Australia). The rats were transferred to a surgical table and the nose placed in a nose cone delivering 3-3.5% isoflurane in 100% oxygen at a flow rate of 0.8 L/min. Surgery was initiated once a deep surgical level of anesthesia was obtained, as evidenced by loss of the pedal withdrawal and corneal reflexes. Body temperature was

maintained at 37.5°C with a heat pad (TC-1000 Temperature controller, CWE Inc, USA) that was covered with an autoclaved absorbent pad. The femoral artery and femoral vein were cannulated for measurement of arterial pressure and drug administration, respectively (PE10 tubing (ID 0.28 x OD 0.61 mm) connected to PE50 (0.7 x 1.45 mm)). For diaphragm electromyography (DiaEMG) recordings, a lateral abdominal incision was made and two nylon-insulated stainless-steel wire electrodes ending with suture pads were placed in the costal diaphragm. Isoflurane was slowly replaced by urethane (1.2 mg/kg i.v., SigmaAldrich, St Louis, USA), following which rats were tracheotomized, and oxygen (100%) directed over the tracheotomy cannula. Rats were placed in a stereotaxic frame with the nose slightly dorso-flexed (mouth bar set at 0). The arterial catheter was connected to a pressure transducer (P23 Db, Statham Gould, USA), and recordings of DiaEMG were amplified and filtered (500-1000 Hz band-pass, NeuroLog System, Digitimer Ltd., UK). Both signals were sampled (5 kHz) and digitized (Power 1401, Cambridge Electrical Design, Cambridge, UK), and monitored using Spike2 software (Cambridge Electrical Design, Cambridge, UK).

Then, for both groups: A midline incision was made over the occipital bone, portions of the bone overlying the cerebellum were removed with a dental drill, and bilateral optical fibers (200 μm diameter; Doric Lenses, Quebec, Canada) connected to a 473nm DPSS Laser (Shanghai Laser & Optics Century Co., Shanghai, China) were descended vertically. Optical fibers were positioned 1.5 mm lateral (lambda level), and descended slowly while photostimulating at 20 Hz (10 ms pulses) until maximal respiratory and bradycardic effects were observed. The experimental protocol was then started after a 15 min period of acclimatization.

## In vitro slice electrophysiology

Rats were deeply anesthetized with 5% isoflurane and the medulla removed, blocked and rapidly cooled in artificial cerebrospinal fluid (aCSF, 2 °C). The medulla was sectioned into 250 μm horizontal slices containing the transduced neurons (Leica VT1200s). Slices were continuously perfused in aCSF solution, containing (in mM): NaCl, 125; KCl, 3; $KH_2PO_4$, 1.2; $MgSO_4$, 1.2; $NaHCO_3$, 25; dextrose, 10; $CaCl_2$, 2; (300 mOsmol), bubbled with 95% $O_2$, 5% $CO_2$, at 32°C. Recording pipettes (3.5 – 4.5 MΩ) contained a low Cl⁻ internal solution (10 mM Cl-) containing (in mM): NaCl, 6; NaOH, 4; KOH, potassium gluconate, 130; EGTA, 11; $CaCl_2$, 1; HEPES, 10; $MgCl_2$, 1; 0.1% biocytin (pH 7.3, 290 mOsmol). As a consequence ECl⁻ was -68.8 mV. Pipettes were visually guided to NTS transduced YFP-neurons using a fixed stage scope (Zeiss Examiner) and camera (Rolera EM-C2, Q-Imaging). Whole-cell recordings were made in either voltage clamp (holding voltage -60 mV or -40 mV) or current clamp mode (MultiClamp 700B and pClamp 10.3, Molecular Devices). Signals were sampled at 20 kHz and filtered at 10 kHz. Liquid junction potentials were not corrected (-6.2 mV at 32°C). Pulses of light (465 nm, 10 mW at fibre tip) from a light emitting diode (LED, Plexon) were delivered via fibre optic affixed to a micro-manipulator and positioned to illuminate the recorded neurons.

## Working Heart-Brainstem preparation

Rats were anesthetized deeply with isoflurane until loss of the pedal withdrawal reflex, bisected below the diaphragm, exsanguinated, cooled in Ringer solution on ice (composition in mM: 125 NaCl, 24 $NaHCO_3$, 5 KCl, 2.5 $CaCl_2$, 1.25 $MgSO_4$, 1.25 $KH_2PO_4$ and 10 dextrose, pH 7.3 after saturation with carbogen gas (chemicals were purchased from Sigma-Aldrich, Australia)) and decerebrated precollicularly. The lungs were removed and the descending aorta was isolated and cleaned. Retrograde perfusion of the thorax and head was achieved via a double-lumen catheter (ø 1.25 mm, DLR-4, Braintree Scientific, Braintree, USA) inserted into the descending aorta. The perfusate was Ringer solution containing Ficoll (1.25%), vasopressin acetate (0.5 nM, V9879, Sigma-Aldrich, Australia) and vecuronium bromide (2–4 μg.ml⁻¹, Organon Teknika, Cambridge, UK) warmed to 31°C and gassed with carbogen (closed loop reperfusion circuit, 25–30 ml/min). The second lumen of the cannula was connected to a transducer to monitor perfusion pressure (PP) in the aorta. The head of the preparation was fixed with ear and mouth bars with the nose ventro-flexed (40°). Simultaneous recordings of phrenic nerve activity (PNA), cervical vagus nerve (left side) activity (VNA) and thoracic sympathetic chain activity (tSNA, between T8-10) were obtained using glass suction electrodes. Some WHBPs for both GtACR2 and ChR2 experiments were performed without VNA recordings, to preserve a fully functional vagal cardiac parasympathetic modulation by keeping both vagus nerves intact. No differences in HR and respiratory sinus arrhythmia were found between WHBPs with either unilateral or

bilateral intact vagus nerves, therefore experiments were pooled for analysis. Neurograms were amplified, filtered (50–1500 Hz band-pass, NeuroLog system, Digitimer Ltd., Welwyn Garden City, UK), rectified and integrated using Spike2 (Cambridge Electrical Design, Cambridge, UK). HR was derived by using a window discriminator to trigger from the R-wave of the electrocardiogram recorded simultaneously through the phrenic nerve electrode. Bilateral optical fibers (200 µm diameter; Doric Lenses, Quebec, Canada) connected to a 473 nm DPSS Laser (Shanghai Laser and Optics Century Co., Shanghai, China) were descended, 20° angled in the rostro-caudal axis, tip pointing forward, to a position 400 µm dorsal to the injection coordinates. Pilot mapping experiments indicated that light delivery to these coordinates was associated with the largest physiological effects.

## Optogenetic stimulation

A light meter (PM20A fiber power meter, Thorlabs, Newton, NJ, USA) was used to calibrate light output from each optical fiber to 10 mW as this was found to induce maximal effects in our experimental set-up without any adverse effect on the tissue (*Menuet et al., 2014*). Photostimulation consisted of: i) frequency-response trials to compare physiological effects induced by *Gt*ACR2 photoactivation in the WHBP (5, 10, 20 and 50 Hz, 10 ms pulses, with respectively 300, 500, 1000 and 1500 pulses at each frequency) and at the single-cell level in vitro (1, 10, 20 and 50 Hz; 1, 10 and 100 ms pulses, 30s each). The results of these trials indicated that maximal physiological effects were evoked by 10 ms light pulses delivered at 20 Hz; in subsequent experiments in vivo or in the WHBP we used tetanic (20 Hz, 300-600 pulses), intermittent (1 Hz, 300 pulses), or single triggered 10ms pulses delivered at specific phases of the respiratory cycle. All photostimulation protocols were repeated multiple times, in a randomized manner, in each animal (technical replication). We found strong intra-preperation reproducibility for all protocols.

## Histology

At the end of WHBP experiments, brainstems were removed, fixed by immersion in 4% formaldehyde in 0.1M sodium phosphate buffer (FA), for 24 hr at 4°C and cryoprotected in 20% sucrose. At the end of the in vivo and tracing experiments, animals were perfused transcardially with 200 ml phosphate-buffered saline followed by 400 ml of 4% FA. Brainstems were removed, post-fixed for 12 hr in 4% FA at 4°C and cryoprotected in 20% sucrose. Brainstems were mostly cut coronally (40 µm sections), except for parvalbumin labeling where they were cut sagitally (50 µm sections), using a cryostat. Fluorescence immunohistochemistry was performed as previously described (*Chen et al., 2010*; *Menuet et al., 2014*; *Sevigny et al., 2012*). Primary antibodies used were rabbit anti-TH (1:5000, Merck-Millipore Bioscience Research Reagents, Bayswater, VIC, Australia, AB152), chicken anti-GFP (1:5000, Abcam, Melbourne, VIC, Australia, AB13970), mouse anti-parvalbumin (1:10,000, Merck-Millipore Bioscience Research Reagents, Bayswater, VIC, Australia, MAB1572), Rabbit anti-NK1R (1:5000, Merck-Sigma-Aldrich, Castle Hill, NSW, Australia S8305), Rabbit anti-DsRed (1:5000, Takara Bio, Mountain View, CA, USA, 632496) and Goat anti-ChAT (1:1000; Merck-Millipore Bioscience Research Reagents, Bayswater, VIC, Australia, AB144P). The secondary antibodies used were Cy3-conjugated donkey anti-rabbit (1:500, Jackson ImmunoResearch Laboratories, West Grove, USA), Cy3-conjugated donkey anti-mouse (1:500, Jackson ImmunoResearch Laboratories, West Grove, USA), AlexaFluor-488 donkey anti-rabbit (1:500, Jackson ImmunoResearch Laboratories, West Grove, USA), AlexaFluor-488 donkey anti-mouse (1:500, Jackson ImmunoResearch Laboratories, West Grove, USA) and AlexaFluor-488 donkey anti-chicken (1:500, Jackson ImmunoResearch Laboratories, West Grove, USA).

In situ hybridization cRNA probes for glycine transporter 2 (GlyT2) and glutamic acid decarboxylase 67 (GAD67) were synthesized as previously described, permitting identification of glycinergic and GABAergic neurons respectively by in situ hybridization (*Bowman et al., 2013*; *Le et al., 2016*). A digoxigenin-labeled cRNA probe for vesicular glutamate transporter 2 (VGluT2), a marker of glutamate synthesis, was synthesized as follows: cDNA template (886 bp, Genebank reference sequence NM_053427) was amplified using the PCR primers (lower case) with T7 and Sp6 RNA polymerase promotors (upper case) attached F:GGATCCATTTAGGTGACACTATAGAAG tcaatgaaatccaacgtcca; R:GAATTCTAATACGACTCACTATAGGGAGAcaagagcacaggacaccaaa.

Following purification of the DNA template with gel extraction, antisense riboprobes were in vitro transcribed (Epicentre Technologies, Madison, WI) incorporating digoxigenin-11-UTP (Roche Applied Science) and validated as previously described (*Kumar et al., 2012*).

Sections containing GtACR2-YFP-labeled neurons were incubated in pre-hybridization buffer (50% formamide, 100 µg/ml heparin, 5 x SSC, pH 7.0, 1 x Denhardt's solution, 250 µg/ml herring sperm DNA, 100 µg/ml yeast tRNA, 5% dextran sulphate, 0.1% Tween-20, reagents obtained from Sigma, Australia unless otherwise indicated) at 37°C (30 min), then 58°C (1 hr) before hybridization with gentle agitation with GlyT2, GAD67 or VGluT2 riboprobe (1000 ng/ml) at 58°C (12–18 hr). Sections were washed in 2 x SSC buffer with 0.1% Tween-20, followed by 0.2 x SSC buffer with 0.1% Tween-20, and then maleic acid buffer (0.1 M maleic acid, 0.15 M NaCl, 0.1% Tween-20). The tissue was then blocked in maleic acid buffer containing 2% Boehringer blocking reagent (Roche Applied Science, Manheim, Germany) and 10% normal horse serum.

Rabbit anti-GFP (1:500, Life Technologies, Scoresby, Australia, A-6455) was added to the blocking buffer and incubated at 4°C (24 hr), then at room temperature (4 hr). Sections were washed in TPBS (Tris–HCl 10 mM, sodium phosphate buffer 10 mM, 0.9% NaCl, pH 7.4, 3 × 30 min) and incubated overnight with AlexaFluor-488 donkey anti-rabbit IgG secondary antibody (1:500, Life Technologies, Scoresby, Australia, A21206) with 2% normal horse serum. DIG-labeled neurons were revealed by incubation in NTMT (0.1 M NaCl, 0.1 M Tris-HCl pH 9.5, 0.1 M MgCl$_2$, 0.1% Tween-20, 2 mM tetramisole HCl) containing nitro blue tetrazolium (Roche Applied Science) and 5-bromo-4-chloro-3-indolyl phosphate salts (Roche Applied Science). The reaction was stopped by washing (0.1M Tris, 1 mM EDTA, pH 8.5, 3 × 15 min) when DIG-labelling was intense with minimal background staining. No labeling was seen when the sense probe was substituted for the anti-sense probe.

## Mapping of reporter expression

A complete map of the distribution of *Gt*ACR2-YFP expression from the 12 WHBP experiments was generated from 1:4 series of sections co-labeled with ChAT, TH and NK1R to assist with alignment of the sections into the Paxinos-Watson atlas (*Paxinos and Watson, 2004*). Heatmaps of pooled *Gt*ACR2-YFP expression, imaged using Zeiss Axio Imager.M2 microscope with an Apocromat 20x/0.8 air objective (Carl-Zeiss, NSW, Australia) and captured using Axiocam 506 mono camera and ZEN 2.6 (blue edition) imaging software, were made using Affinity Designer software (Serif Ltd., UK). Sections from four representative animals used in the WHBP were processed to examine colocalization of preBötC *Gt*ACR2-YFP with GAD67, VGluT2 and GlyT2 mRNA. Sections were imaged under epifluoresence and brightfield (Zeiss Z1, 10x objective, 0.3 NA). The expression profile of ChR2-tdTomato and *Gt*ACR2-YFP used for in vivo experiments were similar to that of *Gt*ACR2-YFP from WHBP experiments, so they were mapped qualitatively rather than quantitatively. The brainstems from two animals from the *Gt*ACR2-YFP in vivo experiments were cut sagitally for better visualization of rostro-caudal *Gt*ACR2-YFP expression.

For the tracing study, tdTomato expression at the injection site was examined under Zeiss Axio Imager.M2 microscope with an Apocromat 20x/0.8 air objective (Carl-Zeiss, NSW, Australia) and captured using Axiocam 506 mono camera and ZEN 2.6 (blue edition) imaging software. TdTomato terminals were imaged with a Zeiss LSM880, Axio Imager 2 confocal microscope, with a Plan-Apochromat 63x/1.4 oil DIC M27 objective, in a Z-stack array, acquiring up to 8 consecutive overlapping slices. All channels were acquired at a consistent pinhole setting of 1.0 µm. Each optical section was assessed for close apposition between axons labelled by tdTomato and ChAT-immunoreactive nucleus ambiguus vagal preganglionic neurons or TH-immunoreactive C1 pre-sympathetic neurons. Maximum intensity projections of 4–6 image stacks were produced using the Image J software (NIH, USA) to better represent the distribution of preBötC terminal labeling.

## Data analysis and statistics

Nerve signals were rectified and integrated with a 50 ms time constant. For low-frequency photostimulation (1 Hz) laser triggered averaging of tSNA and VNA, the signals were rectified and integrated with a 5 ms time constant. For triggered photostimulations, PNA signals were rectified and integrated with a 0.5 ms DC remove time constant and a 100 ms smoothing time constant, to eliminate the ECG background. Phrenic-triggered (end of inspiratory burst) averaging of tSNA, systolic PP and

HR were used to quantify respiratory modulation of tSNA (RespSNA), Traube-Hering waves (mmHg) and respiratory sinus arrhythmia (RSA, bpm), respectively. Tonic levels of tSNA were measured using the phrenic-triggered averaging of tSNA, as the average of tSNA during the expiratory period, therefore outside of the burst of respiratory modulation of SNA. Phrenic-triggered (end of inspiratory burst) averaging of VNA was used to quantify the peak post-inspiratory and non-respiratory components of VNA. Details regarding the period analysed are shown on *Figure 3—figure supplement 2*. For manual photostimulations, to avoid averaging distortion, phrenic-triggered averagings were performed on both consecutive respiratory-related activities to be analysed, and then traces were overlapped and merged at their mean intersection for further analysis. Statistical analysis was done on raw values, and then mean deltas in percentages were quantified (photostimulation vs pre-photostimulation period). Statistics shown on figures presenting only mean delta percentages (for presentation clarity, e.g. *Figure 3*) were done on the raw values as shown on the associated supplementary figures (e.g. *Figure 3—figure supplement 1* and associated data source). Group data are presented as mean ± SEM. Normal distribution of data was verified with the Shapiro-Wilk test, and equal variance of data was tested (SigmaPlot v12, Systat Software Inc, Erkrath, Germany). Data were analyzed using one-way repeated measures or two-way repeated-measures ANOVA, followed by *post hoc* Holm-Sidak multiple-comparison tests, or paired student's *t* tests, for data with normal distribution and equal variance. Data with failed normal distribution or failed equal variance were analyzed using Friedman repeated measures analysis of variance on ranks followed with pairwise multiple comparison Tukey test, or Wilcoxon signed rank test. Linear regression and correlation analysis was used to analyze parameters relationships (Prism v2.0, GraphPad, La Jolla, USA). When missing values were present, for instance because one photostimulation frequency could not be performed or some of the associated data were not appropriate for analysis (noise etc.), it was dealt with accordingly depending on the test used, mostly by excluding the sample from the whole analysis (cf. source data and statistics). All source data and detailed statistics are presented as supplementary information along with each figure. Differences were considered significant at p<0.05.

## Acknowledgements

The authors acknowledge the facilities and technical assistance of the Biological Optical Microscopy Platform (University of Melbourne) for the confocal microscopy images.

## Additional information

### Funding

| Funder | Grant reference number | Author |
|---|---|---|
| National Health and Medical Research Council | App #1120477 | Clément Menuet<br>Simon McMullan<br>Andrew M Allen |
| National Health and Medical Research Council | App #1156727 | Clément Menuet<br>Simon McMullan<br>Andrew M Allen |
| Australian Research Council | DP120100920 | Simon McMullan<br>Andrew M Allen |
| Australian Research Council | DP170104582 | Andrew M Allen |
| University of Melbourne | - McKenzie Research Fellowship | Clément Menuet |
| Fondation pour la Recherche Médicale | Fellowship ARF20160936221 | Clément Menuet |

The funders had no role in study design, data collection and interpretation, or the decision to submit the work for publication.

## Author contributions
Clément Menuet, Andrew M Allen, Conceptualization, Resources, Data curation, Formal analysis, Supervision, Funding acquisition, Validation, Investigation, Visualization, Methodology, Writing - original draft, Project administration, Writing - review and editing; Angela A Connelly, Jaspreet K Bassi, Jessica Kamar, Natasha N Kumar, Data curation, Formal analysis, Investigation, Methodology; Mariana R Melo, Conceptualization, Formal analysis, Investigation, Methodology; Sheng Le, Formal analysis, Investigation, Methodology; Stuart J McDougall, Resources, Data curation, Formal analysis, Investigation, Methodology; Simon McMullan, Resources, Data curation, Formal analysis, Funding acquisition, Investigation, Methodology, Writing - review and editing

## Author ORCIDs
Clément Menuet (iD) https://orcid.org/0000-0002-7419-6427
Stuart J McDougall (iD) http://orcid.org/0000-0002-8778-675X
Andrew M Allen (iD) https://orcid.org/0000-0002-2183-5360

## Ethics
Animal experimentation: Experiments were conducted in accordance with the Australian National Health and Medical Research Council 'Code of Practice for the Care and Use of Animals for Scientific Purposes' and were approved by the University of Melbourne Animal Research Ethics and Biosafety Committees (ethics ID #1413273, #1614009, #1814599 and Florey 16-040).

## Decision letter and Author response
Decision letter https://doi.org/10.7554/eLife.57288.sa1
Author response https://doi.org/10.7554/eLife.57288.sa2

# Additional files
## Supplementary files
• Transparent reporting form

## Data availability
All data generated or analysed during this study are included in the manuscript and supporting files. Source data and statistics files have been provided for Figures 2, 3, 4, 5, 6, 7 and 8.

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
