## [Decision Letter]

Thank you for submitting your article "PreBötzinger complex neurons drive respiratory modulation of blood pressure and heart rate" for consideration by *eLife*. Your article has been reviewed by three peer reviewers, and the evaluation has been overseen by a Reviewing Editor and Ronald Calabrese as the Senior Editor The following individuals involved in review of your submission have agreed to reveal their identity: Patrice Guyenet (Reviewer #1); Davi J. A. Moraes (Reviewer #2); Thomas Dick (Reviewer #3).

The reviewers have discussed the reviews with one another and the Reviewing Editor has drafted this decision to help you prepare a revised submission.

Summary:

The proposed study examines the potential functional connections between the rhythmogenic respiratory network of the preBötzinger complex that drives inspiration and circuits involved in autonomic functions such as those controlling heart rate and blood pressure. On working-heart brainstem preparations and using optogenetics, anatomical investigations and complex measurements of diverse physiological parameters the authors aimed to demonstrate that preBötC neurons directly modulate cardiovascular activity with inhibitory preBötC neurons modulating cardiac parasympathetic activity and excitatory preBötC neurons modulating sympathetic vasomotor activity. Consequently, heart rate and blood pressure oscillate in phase with respiration.

All three reviewers and myself acknowledge the high technical quality of the study performed with the use of sophisticated approaches, nice illustrations and particularly anatomical data, and the presentation of experiments with complex measurements of physiological parameters. This paper addresses the complex question regarding the regulation exerted by the rhythmogenic network of the preBötzinger complex on the autonomic outflow to the vasculature and heart, which is definitively a topic of interest for a large audience. This work undoubtedly brings new findings that will help further identification of the actors of the coupling between respiratory circuits and cardiovascular control, but some concerns (detailed below) must be addressed and revisions on data interpretation should be considered by the authors before publication in *eLife*.

Essential revisions:

1) Throughout the manuscript, the authors should report not just the magnitude of the respiratory peak of SNA but the total SNA (rectified power per unit of time minus background) and where they zero SNA (noise) also. The respiratory modulation is just one input among many. Attenuation of the respiratory modulation is hard to interpret without knowing what happens to the overall signal. For example, the loss of the respiratory modulation means little if the overall signal disappears.

Based on prior evidence in anesthetized rats inhibiting the preBotC region only eliminates the respiratory oscillation of SNA and that of the presympathetic neurons of the RVLM (e.g. Koshiya and Guyenet, 1996). Inhibiting the preBotC region did not reduce the substantial basal activity of these neurons. That is the reason why the authors should report the effects of their optogenetic manipulations on the actual total SNA (rectified signal minus noise /unit of time) and not just the phasic component. This would provide a more realistic estimate of the effects of inhibiting the preBotC region on this outflow and it would help put the phasic component in some perspective.

2) Data interpretation:

The main point of the paper is that the "pre-Botzinger complex", a region of the ventrolateral medulla (VLM) that is required for breathing also drives sympathetic tone to the cardiovascular system. Track tracing experiments complement the evidence by suggesting that this excitatory drive could be mediated via direct projections from the "pre-Botzinger complex" to a more anterior region of the VLM that contains presympathetic bulbospinal neurons. While the data are generally consistent with this hypothesis, the interpretation is based on measures of the output of the system (phrenic nerve discharge (PND) and sympathetic nerve activity (SNA) and the underlying circuit is not clarified. The VLM is extremely complex and, as noted by the authors, the preBotC harbors several types of "respiratory" neurons. This region also contains inhibitory interneurons that mediate the baroreflex (CVLM), adrenergic neurons, cardiovagal neurons etc… The overall effects produced by optogenetic activation of a chloride channel or ChR2 expressed by a subset of these neurons can only be guessed at because the investigators did not perform unit recordings to identify the effect of the light on identified cell types. For example, 50% of prePreBotC are inhibitory neurons. These neurons could very plausibly contribute a phasic respiratory inhibitory input to nearby CVLM GABAergic neurons that innervate RVLM presympathetic neurons and mediate the sympathetic baroreflex. This hypothetical scenario could explain reasonably well the reduction in SNA caused by activating Cl channels expressed by "preBotC" neurons. In other words the present results can be explained by the classic notion of a respiratory gating of the baroreflex at the CVLM level. Clearly many other possibilities could be conceived of but this one should perhaps be given first consideration given that it is the only one that is directly supported by single neuron electrophysiological evidence. So the Discussion should be more careful in its interpretation of the underlying circuitry.

Also: It is not clear why the pre-Bötzinger Complex inhibition induces such sustained and robust bradycardia. The authors discuss the possible contribution of inhibitory tonic neurons in the pre-Bötzinger Complex, which control the firing of cardio-inhibitory parasympathetic neurons in the NA. This data is suggesting that the pre-Bötzinger Complex is controlling a significant component of the heart rate (from 300 to 150 bpm; Figure 3A). On the other side, the pre-Bötzinger Complex excitation did not induce tachycardia. One possibility is that the authors did not manipulate the same population of cells between the experiments (photoinhibition vs. photoexcitation). Therefore, could the authors describe whether the numbers of transfected glutamatergic, GABAergic, and glycinergic neurones were similar between the experiments?

3) An additional trace in a panel for clarification is requested in Figure 8B. Please provide a trace for the average ∫VNA.

4) The present interpretation ignores the role of the pons and plasticity in mediating cardiorespiratory coupling. How can the focus on the Pre-Bötzinger Complex explain the attenuation or even absence of cardio-respiratory coupling with pontine perturbations? How can you explain the variance of the magnitude of cardio-respiratory coupling between elite athletes and the depressed, given that both groups are breathing? There must be multiple influences coupling cardio-respiratory rhythms. Please discuss further the possibility that the respiratory modulation of SNA could come from anywhere including the dorsolateral pons.

---

## [Author Response]

Essential revisions:1) Throughout the manuscript, the authors should report not just the magnitude of the respiratory peak of SNA but the total SNA (rectified power per unit of time minus background) and where they zero SNA (noise) also. The respiratory modulation is just one input among many. Attenuation of the respiratory modulation is hard to interpret without knowing what happens to the overall signal. For example, the loss of the respiratory modulation means little if the overall signal disappears.Based on prior evidence in anesthetized rats inhibiting the preBotC region only eliminates the respiratory oscillation of SNA and that of the presympathetic neurons of the RVLM (e.g. Koshiya and Guyenet, 1996). Inhibiting the preBotC region did not reduce the substantial basal activity of these neurons. That is the reason why the authors should report the effects of their optogenetic manipulations on the actual total SNA (rectified signal minus noise /unit of time) and not just the phasic component. This would provide a more realistic estimate of the effects of inhibiting the preBotC region on this outflow and it would help put the phasic component in some perspective.

We agree with the reviewers that all components of SNA should be analysed, starting with total SNA. We did this, but obviously without sufficient clarity and emphasis. We have generated a new figure, Figure 3—figure supplement 2, that complements the Materials and methods section to explain how SNA was analysed.

In relation to the measurement of total SNA. We analysed the mean integrated tSNA over several periods, as shown on Figure 3—figure supplement 2A. All our analyses were performed using a repeated measures approach within animals, i.e. comparing photostimulation periods relative to intra-preparation pre-photostimulation control and post-photostimulation recovery periods. When comparing in this way it is not necessary to remove noise in the recordings before analysis, since it will be automatically discarded by the repeated measures analysis process. For each animal, noise is reasonably expected to be the same in each period of SNA analysis within each preparation. When measuring noise at the completion of an experiment, and then extracting it from the analysed record, it is assumed that the noise remained constant throughout, but this is less likely to be true and a comparison at relatively close experimental periods will be more accurate. Removing noise is only necessary for comparing absolute values between preparations, which we didn’t do here.

Using this approach, we analysed total SNA for both the photoinhibition and photoexcitation protocols. The SNA data included in Figure 3 only interrogates the effect of prolonged preBötC photoinhibition on total SNA. Then, we performed phrenic triggered averaging of tSNA to analyse specifically the respiratory modulation of tSNA, which we called RespSNA to highlight the difference with total SNA, and to analyse the non-respiratory component of tSNA, which we called tonic tSNA (now explained in Figure 3—figure supplement 2B). Figure 6B shows the analysis of RespSNA and tonic SNA which demonstrates little change in tonic SNA and substantial reduction in the RespSNA. We made this detailed analysis with the same reasoning as expressed by the reviewers, to enable interpretation of the RespSNA changes compared to non-respiratory, tonic tSNA and compared to total tSNA.

We thank the reviewers for mentioning the study by Koshiya and Guyenet, 1996, which suggests similar conclusions to our present study. It should be noted that in the cited study, muscimol injection in the preBötC area removes the respiratory component of SNA, but it is concomitant with total arrest of the respiratory nerves recorded. It is therefore difficult to conclude from this study, if the respiratory component of SNA is removed directly because of the inhibition of the preBötC area, or indirectly because of the global respiratory arrest. We now discuss this in the subsection “The preBötC provides excitatory drive to vasomotor sympathetic activity”.

2) Data interpretation:The main point of the paper is that the "pre-Botzinger complex", a region of the ventrolateral medulla (VLM) that is required for breathing also drives sympathetic tone to the cardiovascular system. Track tracing experiments complement the evidence by suggesting that this excitatory drive could be mediated via direct projections from the "pre-Botzinger complex" to a more anterior region of the VLM that contains presympathetic bulbospinal neurons. While the data are generally consistent with this hypothesis, the interpretation is based on measures of the output of the system (phrenic nerve discharge (PND) and sympathetic nerve activity (SNA) and the underlying circuit is not clarified. The VLM is extremely complex and, as noted by the authors, the preBotC harbors several types of "respiratory" neurons. This region also contains inhibitory interneurons that mediate the baroreflex (CVLM), adrenergic neurons, cardiovagal neurons etc… The overall effects produced by optogenetic activation of a chloride channel or ChR2 expressed by a subset of these neurons can only be guessed at because the investigators did not perform unit recordings to identify the effect of the light on identified cell types. For example, 50% of prePreBotC are inhibitory neurons. These neurons could very plausibly contribute a phasic respiratory inhibitory input to nearby CVLM GABAergic neurons that innervate RVLM presympathetic neurons and mediate the sympathetic baroreflex. This hypothetical scenario could explain reasonably well the reduction in SNA caused by activating Cl channels expressed by "preBotC" neurons. In other words the present results can be explained by the classic notion of a respiratory gating of the baroreflex at the CVLM level. Clearly many other possibilities could be conceived of but this one should perhaps be given first consideration given that it is the only one that is directly supported by single neuron electrophysiological evidence. So the Discussion should be more careful in its interpretation of the underlying circuitry.

The complexity of this region, and our intentional choice of viral constructs to target all preBötC neurons, resulted in expression of the opsins in various types of neurons. This could indeed lead to multiple neuronal pathways being affected during photostimulations. We have altered the Discussion accordingly, subsection “The preBötC provides excitatory drive to vasomotor sympathetic activity”, including the preBötC-CVLM-RVLM hypothesis. Still, the preBötC-RVLM connectivity is robust, as supported by both functional and anatomical evidence. C1 neurons show inspiratory-phase excitatory post-synaptic potentials (Moraes et al., 2013 ). There is mono-synaptic connection between NK1-R positive preBötC neurons and RVLM bulbo-spinal neurons including C1 neurons (Dempsey et al., 2017, and Menuet et al., 2017).

Also: It is not clear why the pre-Bötzinger Complex inhibition induces such sustained and robust bradycardia. The authors discuss the possible contribution of inhibitory tonic neurons in the pre-Bötzinger Complex, which control the firing of cardio-inhibitory parasympathetic neurons in the NA. This data is suggesting that the pre-Bötzinger Complex is controlling a significant component of the heart rate (from 300 to 150 bpm; Figure 3A). On the other side, the pre-Bötzinger Complex excitation did not induce tachycardia. One possibility is that the authors did not manipulate the same population of cells between the experiments (photoinhibition vs. photoexcitation). Therefore, could the authors describe whether the numbers of transfected glutamatergic, GABAergic, and glycinergic neurones were similar between the experiments?

Unfortunately, we didn’t do in situ hybridization experiments on the brainstem sections of ChR2 expressing animals, therefore we couldn’t compare expression of the different neurochemical phenotypes expressing ChR2 versus GtACR2. Still, macroscopically, the same area was transfected for both ChR2 and GtACR2 sets of experiments, and the same promoter was used, so it is highly likely that the same profile of cell transduction would be observed between both sets of experiments.

Still, the point raised by the reviewers on why preBötC photoexcitation didn’t produce tachycardia is indeed interesting. Our interpretation is that it reflects the ongoing level of cardiac parasympathetic activity in the working heart-brainstem model. Examination of the response to atropine (Figure 4C) reveals a very small and variable tachycardia, with a mean change of ~15 bpm across the group. If we consider the simplest circuitry, involving a direct respiratory phasic inhibitory output from the preBötC to the cardiac parasympathetic preganglionic neurons, it is not surprising that excitation of these preBötC neurons has little effect. Further inhibition of parasympathetic activity, due to excitation of these inhibitory preBötC neurons, would have little ability to produce further inhibition or tachycardia. This is also consistent with the loss of RSA during preBötC excitation. In contrast, removal of this inhibition, with photoinhibition of the putative inhibitory preBötC neurons, has the potential to induce large changes in heart rate.

3) An additional trace in a panel for clarification is requested in Figure 8B. Please provide a trace for the average ∫VNA.

We have added the average ∫VNA on Figure 8B.

4) The present interpretation ignores the role of the pons and plasticity in mediating cardiorespiratory coupling. How can the focus on the Pre-Bötzinger Complex explain the attenuation or even absence of cardio-respiratory coupling with pontine perturbations? How can you explain the variance of the magnitude of cardio-respiratory coupling between elite athletes and the depressed, given that both groups are breathing? There must be multiple influences coupling cardio-respiratory rhythms. Please discuss further the possibility that the respiratory modulation of SNA could come from anywhere including the dorsolateral pons.

We agree with the reviewers that there are likely multiple influences on cardio-respiratory rhythms. We have chosen to focus on the input from one region where our tracing studies demonstrate a direct, monosynaptic input to RVLM presympathetic and RVLM C1 neurons (Dempsey et al., 2017, and Menuet et al., 2017). Our interest is in the inspiratory modulation of SNA, which is the key component that is amplified in the Spontaneously Hypertensive Rat (SHR). This component is directed through C1 neurons, and removal of these C1 neurons during development substantially reduces the blood pressure of the adult SHR. A prominent source of inspiratory activity in the brain is the Pre-Bötzinger Complex and thus the focus of this study.

In response to the direct comment about the dorsolateral pons, examination of the data indicates that interruption of the pons alters respiratory drive (for a study using the same in situ perfused rat preparation, see Baekey et al., 2008). The problem we have as a field of researchers is in interpreting whether the observed alteration in respiratory-cardiovascular modulation is direct, i.e. the pons directly influences the RVLM pre-sympathetic neurons or the cardiac parasympathetic preganglionic neurons, or, is a result of the altered state of the respiratory network. The key advantage of using photoinhibition, as applied in this study, is being able to apply single pulse stimulation (Figure 5) and observe the response in the absence of any changes to network dynamics. We show a clear effect of the preBötC. We make no attempt to provide an over-arching conclusion in relation to how other brain regions impact cardio-respiratory coupling. In response to the helpful suggestions of the reviewers, we have added to the Discussion, in an attempt to cover some understanding of how other brain regions might impact cardio-respiratory coupling. However with space limitations, and not wishing to over-speculate on the basis of our direct data, this discussion cannot be comprehensive.

Regarding the variance of the magnitude of cardio-respiratory coupling in elite athletes, or in diseased people (hypertension, chronic heart failure etc.), we believe one must differentiate between mechanisms generating cardio-respiratory coupling, and mechanisms regulating it. Indeed, it is likely that external regulation of the gain of the respiratory-sympathetic and respiratory-parasympathetic connections is causing most of the variance in cardio-respiratory coupling magnitude. For example, in rats submitted to chronic intermittent hypoxia, adrenomedullin was shown to induce long-term potentiation of sympathetic vasomotor activity and specifically increase respiratory modulation of SNA (Zoccal et al., Exp Physiol, DOI: 10.1113/EP087613). On the parasympathetic side, Dr Mendelowitz and his lab have produced several studies demonstrating that nicotine enhances neurotransmission to cardiac parasympathetic neurons, including inspiratory GABAergic inputs resulting in enhanced respiratory sinus arrhythmia (Neff et al., 2003). Our study doesn’t investigate the mechanisms regulating cardio-respiratory coupling magnitude, it investigates one specific mechanism generating it. We therefore prefer not to discuss these aspects.